# SinoMultiAffect: A Chinese Multi-Label and Fine-Grained Emotional Text Dataset with fMRI Data

## Abstract

Emotion plays an indispensable role in advancing human-AI interaction, yet the field still lacks high-quality, fine-grained Chinese datasets that integrate both language and neural modalities. We present **SinoMultiAffect** (SMA), a multi-modal emotion dataset designed to advance research on emotion, language and the emotion-related capabilities of artificial intelligence. The dataset consists of 4,500 Chinese sentences in total collected from social media platforms in China, with 4,058 of them labeled with a fine-grained taxonomy of 35 emotion categories (including Neutral) with their intensity, as well as continuous annotations along the valence-arousal-dominance (VAD) dimensions. Our dataset also includes functional magnetic resonance imaging (fMRI) recordings of the brain while human participants were reading the sampled sentences. The utility of the dataset was demonstrated by the predictive performance of large language models (LLMs) on multi-label emotion recognition. We further conduct an exploratory, proof-of-concept analysis of a VAD-guided human-LLM alignment framework, suggesting that incorporating emotional information may enhance the alignment between text and brain embeddings and improve downstream bidirectional retrieval performance. By integrating text, categorical, dimensional, and neuroimaging information, SMA provides a unique resource for studies on emotion and language, offering new opportunities for interdisciplinary research in natural language processing, affective computing, and cognitive neuroscience.

## 1 Introduction

The indispensable role of **EMOTION** in biological development and learning has been extensively examined, showing that it profoundly shapes organisms' behaviour and decision-making (Damasio, 2006; LeDoux, 2013; Tyng et al., 2017). In recent years, with the rapid advancement of artificial intelligence (AI), researchers have explored integrating emotion into AI to enhance affective state recognition, emotion understanding, and the generation of contextually appropriate responses (Hong et al., 2025; Zhang et al., 2023; Schlegel et al., 2025), thereby facilitating more natural human-AI interaction (Amershi et al., 2019; Glickman & Sharot, 2025).

Psychological theories of human emotion suggest that the formation of emotion concepts is insepa-rable from language (Bamberg, 1997; Barrett et al., 2007). Recent evidence suggests that language itself carries structured representations of emotion concepts, which can causally support emotion inference beyond mere communication functions (Li et al., 2024). Existing research on emotion analysis in language typically focuses on how people convey emotions through linguistic means (Buechel & Hahn, 2022). Emotional language is often categorized into two types: expressive (e.g, '*Wow!*') and descriptive (e.g, '*I feel so happy!*') (Kövecses, 2003; Lee et al., 2018), both of which are frequently observed in social media, a major source of daily language emotional expression.

Notably, emotions are inherently shaped by cultural contexts (Lindquist et al., 2022). Prior studies have revealed substantial cross-cultural variation in emotion semantics (Jackson et al., 2019; Cowen & Keltner, 2021), showing that emotion concepts exhibit distinct patterns of association across lan-guage families. Beyond linguistic and cultural variation, emotion concepts are also grounded in em-bodied experience, with the brain constructing such meaning through neural representations shaped

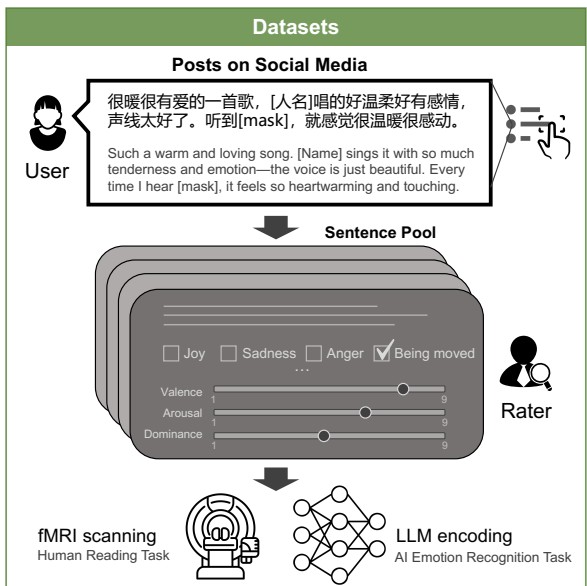

| Datasets | Labels |
|---|---|

**Posts on Social Media**

User: 很暖很有爱的一首歌，[人名]唱的好温柔好有感情，声线太好了。听到[mask]，就感觉很温暖很感动。

Such a warm and loving song. [Name] sings it with so much tenderness and emotion—the voice is just beautiful. Every time I hear [mask], it feels so heartwarming and touching.

**Sentence Pool**

☐ Joy  ☐ Sadness  ☐ Anger  ☑ Being moved

Valence 1 ——————9
Arousal 1 ——————9
Dominance 1 ——————9

Rater

fMRI scanning — Human Reading Task
LLM encoding — AI Emotion Recognition Task

| Polarity | Emotion Labels |
|---|---|
| Positive 🙂 | joy (喜悦), love (爱), looking forward (期待), pride in oneself (得意), optimism (乐观), appreciation(欣赏), satisfaction (满足), trust (信任), sympathy (同情), being at ease (惬意), healing (治愈), gratitude (感激), being moved (感动), admiration (羡慕) |
| Negative 🙁 | anger (愤怒), sadness (悲伤), disgust (厌恶); fear (害怕), disappointment (失望), remorse (懊悔), pity (遗憾), guilt (内疚), envy (妒忌), anxiety (焦虑), embarrassment (尴尬), shyness (害羞), cynicism (愤世嫉俗), pessimism (悲观) |
| Others 😯 | surprise (惊讶), curiosity (好奇), alarm (警觉), confusion (困惑), hesitant (犹豫), nostalgia (怀旧), Neutral (中性) |

- **Intensity:** The strength or magnitude of the chosen emotions.
- **Valence:** The pleasantness of the emotion, ranging from negative and unpleasant to positive and pleasant.
- **Arousal:** The level of activation or excitement, ranging from calm to highly excited.
- **Dominance:** The sense of control expressed in the emotion, ranging from passive or submissive to active, with a desire to influence or dominate others or the environment.

Figure 1: **Dataset overview.** Texts collection and annotation pipeline. The annotated texts are used in both fMRI scanning (human reading task) and LLM-based emotion recognition task **(Left)**. Emotion labels are grouped by polarity into positive, negative and others, along with definitions of the emotion dimensions **(Right)**.

by past interactions and conceptual knowledge (Baldassano et al., 2017; Barrett, 2017a;b). Although datasets such as GoEmotions (Demszky et al., 2020) have significantly advanced fine-grained emotion recognition in English, there remains a growing demand for carefully constructed resources that reflect non-Western culture and include the neural representations of emotion, especially for developing more generalized foundation models that incorporate not only world knowledge but also emotion knowledge.

To this end, we introduce **SinoMultiAffect (SMA)**, a dataset of 4,500 Chinese emotional texts with multi-label and fine-grained annotations and accompanying fMRI data from a human reading task that captures neural representation of emotional processing of texts. Building on prior work, SMA pushes fine-grained emotion categorization further by providing 35 emotion categories, substantially extending the granularity beyond existing datasets, together with continuous ratings along valence, arousal, dominance, and intensity dimensions. The dataset was systematically curated through a rigorous annotation protocol. An overview of the dataset is shown in Figure 1, illustrating the dataset pipeline, the 35 specific fine-grained emotion labels, and the definitions of the emotion dimensions. Because the formation of emotion concepts is inseparable from embodied experience, we collected fMRI data to capture the neural representations of emotional processing. Building on this resource, we designed a framework to align textual and neural representations, guided by valence, arousal, and dominance dimensions, and trained with a bidirectional multi-positive contrastive objective to support brain–text cross-modal retrieval. **Our contributions are summarized as follows:**

- **The SinoMultiAffect dataset:** We introduce SMA, a Chinese multi-modal dataset of 4,500 emotional texts with fine-grained multi-label annotations and accompanying fMRI recordings. The dataset provides 35 categorical emotion labels together with valence, arousal, dominance, and intensity ratings, supporting both categorical and dimensional analyses.

- **Comprehensive zero-shot evaluation:** We assess the ability of 13 open-source large language models to handle a wide spectrum of emotion categories in a zero-shot setting, providing insights into their capacity for fine-level emotion understanding.

- **VAD-guided cross-modal alignment:** We conduct an exploratory, proof-of-concept analysis of a VAD-guided framework for linking textual and neural representations, aiming to investigate whether emotional dimensions contribute to improved cross-modal alignment.

Table 1: Comparison of selected Chinese emotion datasets.

| Dataset | Size | Label (#num) | Modals | Manual | Dim |
|---|---|---|---|---|---|
| **SinoMultiAffect (Ours, 2025)** | 4,500 posts | multi (35) | fMRI | ✔ | ✔ |
| CMACD (Zhou et al., 2024a) | 566,900 posts | multi (6) | ✗ | ✗ | ✗ |
| EmotionTalk (Sun et al., 2025) | 19,250 utterances | single (7) | Audio | ✔ | ✔ |
| CH-MEAD (Ruan et al., 2023) | 25,292 utterances | single (26) | Video&Audio | ✔ | ✗ |
| ResEmo (Hu et al., 2024) | 3,813 posts | single (16) | ✗ | ✔ | ✗ |
| ChineseEmoBank (Lee et al., 2022) | 2,969 texts | - | ✗ | ✔ | ✔ |

Note: Manual = Manual Annotations and Dim = Dimensions.

## 2 RELATED WORK

**Emotion Datasets.** A milestone in textual emotion multi-label datasets is the GoEmotions corpus (Demszky et al., 2020), which contains 58k Reddit comments annotated with 27 fine-grained emotions plus a neutral label. Each sample is manually multi-labeled, making it the largest and most detailed English dataset and a widely used benchmark for nuanced emotion recognition in large language models.

For Chinese datasets, the largest is the Chinese Multi-label Affective Computing Dataset (CMACD) (Zhou et al., 2024a), which consists of 566k Weibo posts annotated with six classical emotion categories. Another notable resource is CH-MEAD (Ruan et al., 2023), a multimodal conversational dataset of 25k utterances designed for chatbot training, where each sample is single-labeled based on multimodal cues. In addition, EmotionTalk (Sun et al., 2025) contains 19k conversational utterances (about 23.6h) from 19 actors, each annotated with a single label out of seven utterance-level emotion categories, confidence score, polarity ratings, and four speech-related dimensions, offering a multimodal resource for dialogue emotion analysis. ResEmo (Hu et al., 2024) consists of 3,813 Weibo posts annotated with 16 single-label emotions, providing a resource for fine-grained Chinese emotion recognition. ChineseEmoBank (Lee et al., 2022) includes 2,969 texts annotated with valence, arousal, and dominance ratings, offering a dimensional perspective on Chinese emotional language. The comparison of the above datasets with ours is presented in Table 1

**LLM-Human Brain Alignment.** Recent research has increasingly examined the alignment between human neural representations of language and the embeddings derived from large language models (LLMs). Contextual embeddings, continuous vector representations of natural language produced by deep language models, have been shown to exhibit common geometric structures with neural embeddings in the human brain (Goldstein et al., 2024). Extending this perspective to development, Evanson et al. (2025) demonstrated that LLMs spontaneously captured the neurodevelopmental trajectory during the training process and learned representations that can only be identified in the adult human brain. This finding suggested that LLMs can serve as a tool that simulates the neurodevelopmental process of language representation from childhood to adulthood. Nevertheless, alignment remains incomplete, Zhou et al. (2024b) uncovered divergences between human brains and LLMs, particularly in domains requiring social–emotional intelligence and physical commonsense, though targeted fine-tuning could partially bridge this gap. Likewise, by analyzing sparse autoencoder features in LLMs, (Li et al., 2025) revealed that both human and artificial systems rely on distributed conceptual geometries, but differ in their abstraction and sparsity principles. More recently, AlKhamissi et al. (2025) systematically analyzed LLM–brain alignment across training, showing that alignment in the language network is primarily driven by formal linguistic competence and saturates early, while functional competence develops later with weaker alignment.

## 3 SINOMULTIAFFECT DATASET

Our dataset consists of 4,500 emotional sentences in total, collected from Chinese social media platforms, where emotion categories are assigned as the consensus label for a sentence only when it is selected by at least two raters. Based on this criterion, 4,058 are labeled with one or more of 34 emotion(s) with their intensity or Neutral, as well as emotion dimensions for each sentence, including valence, arousal, and dominance.

## 3.1 SENTENCE COLLECTION AND CURATION

In order to ensure the diversity of emotion categories, sentences were extracted from various sections of social media, including music, movies, books, topic discussions, online reading communities, real-time feeds, humor, home & lifestyle, and parenting. Preprocessing steps of original texts included removing empty samples, restricting sentence length to 5–50 characters, filtering out sentences with excessive (>50%) of non-Chinese characters or symbols, and deleting samples in which the title of a work occupies more than 80% of the text. Subsequently, directional references such as personal names and work titles were masked, offensive and adult content was filtered using Qwen-Plus (Qwen et al., 2025)[1], and finally, all texts were manually reviewed to ensure data quality and the removal of private information.

## 3.2 ANNOTATION DESIGN

Text annotations were performed according to discrete emotion categories (e.g., joy, anger, sadness) and continuous dimensions (e.g., valence, arousal, dominance) (Ekman et al., 1999; Russell & Mehrabian, 1977). Specifically, for each text sample, annotators were asked to assign one or two discrete emotion categories as well as to provide ratings along the dimensional scales. This dual-labeling scheme enables the dataset to support analyses from both categorical and dimensional perspectives, and facilitates research that bridges the two theoretical perspectives.

The emotion taxonomy in our dataset was informed by prior curation studies. While we drew on categories commonly used in existing datasets, we also incorporated emotions that are particularly salient in East Asian cultures, such as being moved (Tokaji, 2003), which frequently appears in linguistic expressions. A complete description of all emotion categories employed in this dataset is provided in Appendix A.1. Besides, raters were encouraged to leave comments in the note box whenever they felt that the provided labels did not adequately capture the emotion expressed in a sentence, and their suggestions were taken into consideration when appropriate.

## 3.3 ANNOTATION PROCEDURE

Every rater was assigned a random set of 500 sentences from the sample pool, and each sentence was rated by three different participants to achieve consensus.

**Requirements and Instructions.** Participants must be native Chinese speakers and reside in China most of the time, with preference given to those with a background in the humanities or social sciences, or with a strong interest in emotion and language.

**Annotation Platform.** Considering the multiple labels for each text sample, we developed a specialized annotation website for the raters[2]. Rating instructions are available from the website, and each labeling task is accompanied by clear guidelines to ensure that raters can fully understand the requirements.

## 4 DATASET ANALYSIS

Thorough analyses were conducted to verify the quality of emotion labels. The summary statistics of the dataset are presented in Table 2. Sentences without a consensus label were excluded from further verification analysis due to potentially imprecise labeling or ambiguous interpretation. Two emotions (envy and shyness) contain only two sentences with shared labels, so they were removed from subsequent quantitative analyses due to insufficient sample size. All original labeling results and dataset samples are provided in the supplementary materials.

## 4.1 ANALYSIS OF INTERRATER CONSISTENCY

For multi-label recognition, we used the Jaccard index to measure the agreement between raters of each sample. The Jaccard index is defined as the size of the intersection divided by the size of the

---

[1] https://github.com/QwenLM/Qwen
[2] https://sino-multi-affect-6egyxo42cb3818-1373867889.tcloudbaseapp.com/

Table 2: Dataset statistics.

| Number of total examples | 4500 |
|---|---|
| Number of examples with consensus labels | 4058 (90.17%) |
| Number of emotion labels | 35 (including Neutral) |
| Number of unique raters | 27 |
| Number of raters per example | 3 |
| Number of labels per example | 1: 86.67% 2: 11.67% 3: 1.67% |
| Dimension labels | Intensity, Valence, Arousal, Dominance |

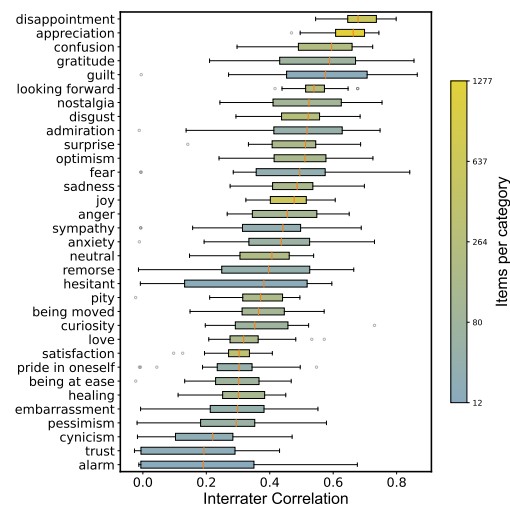

Figure 2: Interrater correlation.

union of two label sets (Real & Vargas, 1996). For each sentence, we calculated the pairwise Jaccard index between all raters and averaged these values to obtain an overall interrater consistency score. The average Jaccard index across all sentences is 0.346, indicating moderate agreement among raters. In addition to sentence-level consistency, we analyzed interrater agreement for each emotion (Delgado & Tibau, 2019) in Figure 2. Specifically, *disappointment*, *appreciation* and *confusion* have the highest interrater correlation, while *cynicism*, *trust* and *alarm* are the lowest. Detailed formula of how to calculate these metrics and results can be found in Appendix B.

## 4.2 VALIDATION OF EMOTION TAXONOMY

In order to verify the current emotion taxonomy and examine the correlation and distinctiveness among emotions, we calculated the Pearson correlation coefficients between every two categories. Specifically, for a dataset with N samples, the binary labels (1 = selected, 0 = not selected) assigned by multiple raters were averaged for each sample to obtain its score on a given emotion (e.g., if three raters label a sample as [1, 1, 0], the averaged score for that emotion is 2/3). Repeating this procedure across all N samples yields an N-dimensional vector for each emotion. Pairwise Pearson correlations between these vectors were then computed to quantify the relationships between emotion categories. Independent sample t-test analysis showed that emotions within the same polarity tend to be more strongly related ($p < 0.001$).

To further illustrate the relationship among these fine-grained emotions, for each category we identified both the most similar and the most distinctive counterparts. To be specific, the dark solid line indicates a positive correlation between two emotions, while the light dashed line demonstrates dissimilarity between emotions. Notably, statistical analysis also showed that negative emotion pairs are more similar than positive emotion pairs ($p < 0.001$), suggesting that negative emotions tend to cluster more closely in semantic space, whereas positive emotions appear more dispersed. Figure 3 illustrates the relationships among fine-grained emotions. To address the possibility of semantic overlap among highly fine-grained categories, we additionally organized the 35 emotions into 17 broader clusters following an established emotion ontology (Xu et al., 2008). This hierarchical structure offers researchers a more flexible representation that can be adapted to diverse analytical needs, while still preserving the nuanced distinctions that motivated the original annotation scheme. The full mapping between the fine-grained and coarse-grained categories is provided in Appendix A.2.

## 4.3 VAD AND EMOTION INTENSITY

Average valence, arousal and dominance of each emotion category as well as the color mapping of intensity in a 3D VAD space are shown in Figure 4. Our analysis showed that emotion intensity in the dataset is most strongly predicted by arousal ($r = 0.41$), with valence ($r = 0.24$) and dominance ($r = 0.18$) playing smaller roles. Positive emotions (e.g., *joy*, *appreciation*) tend to have higher intensity

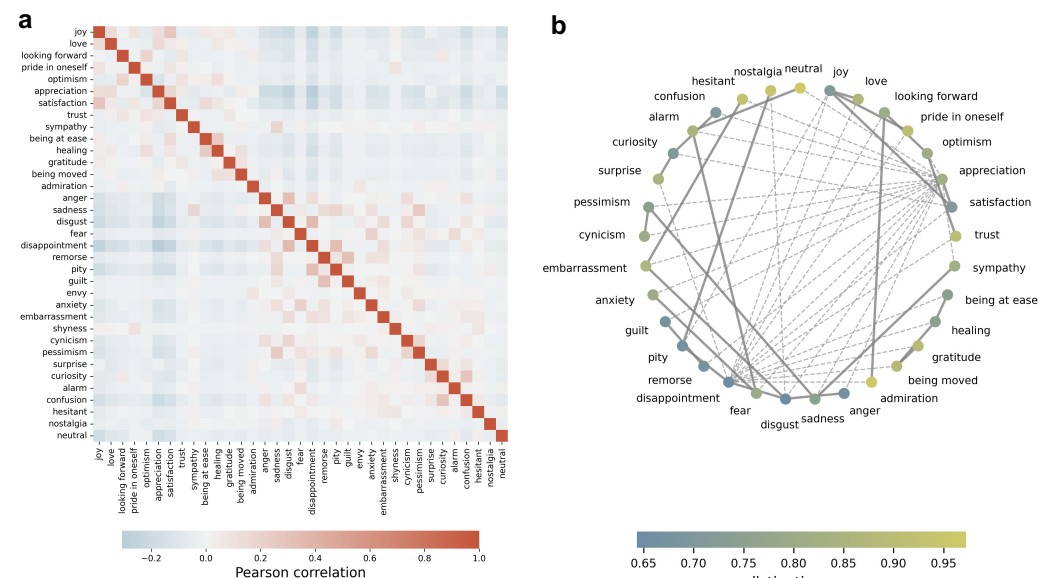

Figure 3: **Emotions relationship and distinctiveness. (a)** Pearson correlation between paired emotions, showing stronger associations within the same polarity (e.g., *joy* and *love* among positive emotions, *anger* and *disgust* among negative emotions). **(b)** For each emotion, the most similar and the most distinctive counterparts are presented.

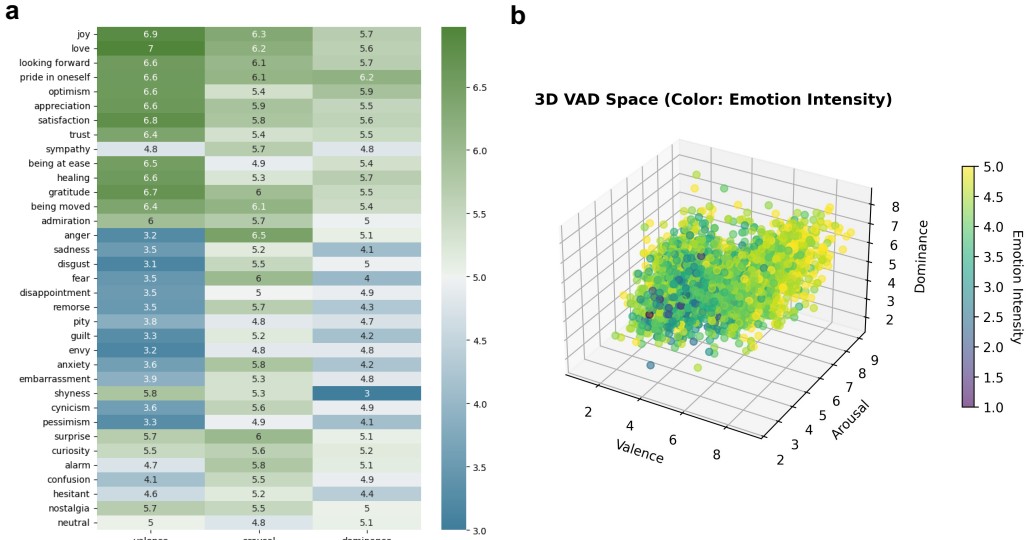

Figure 4: **VAD and emotion intensity. (a)** Average valence, arousal, and dominance scores for each emotion category. **(b)** Distribution of emotions in a 3D VAD space with color indicating intensity.

and show strong positive correlations between valence and arousal, while negative emotions (e.g., *disappointment*, *disgust*) often show negative or weak valence-arousal correlations. Overall, VAD dimensions are moderately correlated but can be treated as relatively independent for modeling purposes.

### 4.4 FMRI DATA DURING READING TASK

To further contribute to the studies on emotional text representation of both human and artificial intelligence, we also sampled 120 sentences from SinoMultiAffect. The sub-dataset was reviewed

by two professional researchers in psychology of emotion. Three participants (all females, mean age = 26 ± 2.16 years) completed the reading task, and all data underwent preprocessing. The detailed information of the fMRI scanning procedure can be found in Appendix C.

## 5 EVALUATION AND HUMAN–LLM ALIGNMENT

### 5.1 DATASET-BASED EVALUATION OF EXISTING LARGE LANGUAGE MODELS

After developing the dataset, we evaluated it on several widely used open-source large models. This serves three purposes: (i) to validate the effectiveness and quality of the dataset; (ii) to establish strong baselines for future research; and (iii) to ensure comparability with existing studies by leveraging commonly adopted foundation models.

#### 5.1.1 EXPERIMENTAL SETUP

**Models.** We evaluated 13 open-source LLMs spanning 8B–72B parameters and diverse architectures, covering major families such as Llama-3.1, Gemma-3, GLM-4, Mistral, and Qwen-2.5.

**Prompting Strategy.** A zero-shot Chinese prompt was designed, presenting the full label set within a unified multiple-choice format. Each emotion label is mapped to a unique symbolic key, utilizing the standard Latin alphabet ($A - Z$) extended with Greek letters ($\alpha, \beta, \dots$) since the number of Latin letters is not enough. The model is explicitly instructed to analyze the semantic content of the input text and predict all applicable emotions by selecting their corresponding keys (see detailed in appendix D.1).

**Evaluation Metrics.** Performance was assessed using Jaccard Similarity along with Precision, Recall and Exact Match Accuracy (formulas in Appendix D.2). To effectively handle multi-label classification, we employ the *Choice Funnel* algorithm (Xu et al., 2025). This method iteratively selects the option with the highest first-token probability and removes it from the candidate set. The process terminates when the "None of the above" option is selected or the prediction confidence falls below a predefined threshold. In addition, we extracted the last-token embeddings of each sentence from all models, which provide a consistent representation of the model's overall emotional interpretation. Besides, using the human-annotated emotion with the highest intensity as a reference, we examined several representation-level properties of each model, including the correlation between model-derived and human emotion representations, the distinctiveness of fine-grained emotion clusters, and the predictive accuracy on dimensional ratings.

#### 5.1.2 PERFORMANCE OF DIFFERENT MODELS

The overall performance of the evaluated models is presented in Table 3. All evaluated models performed better than a random chance baseline, which yielded an F1-score close to zero, confirming that the models can effectively generalize their pre-trained knowledge to our emotion recognition task.

Qwen2.5-72B shows a sensitivity-oriented pattern with the highest Recall and Jaccard, capturing a broader range of emotions. In contrast, DeepSeek-R1 adopts a more conservative strategy, achieving the highest Precision and Exact Match, indicating more selective but highly reliable predictions. Interestingly, for metrics related to human ratings, we observe that models with higher category distinctiveness, that is, more separable fine-grained emotion clusters in their embedding space, tend to achieve higher Jaccard similarity on the multi-label prediction task. In contrast, greater similarity to human emotion-space structure (Correlation with Human) does not consistently correspond to better multi-label accuracy across models. Detailed results and extended performance analyses are provided in Appendix D.3.

### 5.2 HUMAN–LLM ALIGNMENT WITH VAD-GUIDED REPRESENTATIONS

The evaluation of the performance showed that the existing open-source LLMs with more parameters are more capable of handling fine-grained emotion recognition tasks. Next, we investigated how the dataset with emotion dimension labels can support cross-modal alignment between human brain activity and large language models (LLMs). In particular, we leveraged the continuous

Table 3: Performances of Large Model on Multi-label Emotion Recognition Task

| Model | Jaccard | Prec. | Recall | EM | Human Corr. | Category Dist. | VAD avg |
|---|---|---|---|---|---|---|---|
| DeepSeek-R1-Distill-Llama-70B 2025 | 0.451 | **0.439** | 0.467 | **0.239** | 0.519 | -0.009 | 0.424 |
| Gemma-3-27B-it 2025 | 0.388 | 0.350 | 0.509 | 0.105 | **0.651** | -0.033 | 0.474 |
| GLM-4-9B-Chat 2024 | 0.262 | 0.366 | 0.369 | 0.051 | 0.619 | -0.040 | 0.451 |
| InternLM3-8B-Instruct 2024 | 0.359 | 0.365 | 0.458 | 0.090 | 0.645 | -0.006 | 0.471 |
| Llama-3.1-8B-Instruct 2024 | 0.374 | 0.409 | 0.376 | 0.219 | 0.588 | -0.026 | 0.420 |
| Llama-3.3-70B-Instruct 2024 | 0.397 | 0.385 | 0.547 | 0.103 | 0.639 | -0.003 | 0.438 |
| Mistral-7B-Instruct-v0.3 2023 | 0.287 | 0.308 | 0.404 | 0.058 | 0.567 | -0.049 | 0.389 |
| Mixtral-8x22B-Instruct-v0.1 2024 | 0.378 | 0.320 | 0.500 | 0.102 | 0.556 | -0.030 | 0.414 |
| Qwen2.5-7B-Instruct 2025 | 0.344 | 0.368 | 0.410 | 0.149 | 0.617 | -0.025 | 0.463 |
| Qwen2.5-32B-Instruct 2025 | 0.434 | 0.382 | 0.574 | 0.147 | 0.543 | 0.010 | **0.481** |
| Qwen2.5-72B-Instruct 2025 | **0.453** | 0.410 | **0.583** | 0.163 | 0.572 | **0.012** | 0.478 |
| Qwen3-8B 2025 | 0.390 | 0.398 | 0.533 | 0.125 | 0.569 | -0.015 | 0.461 |
| Qwen3-14B 2025 | 0.396 | 0.374 | 0.507 | 0.123 | 0.453 | -0.026 | 0.432 |

Note: Prec is abbreviation for Precision, EM is abbreviation for Exact Match, Human Corr. is abbreviation for Correlation with Human, Category Dist. is abbreviation for average Distance between paired Emotions, VAD avg. is average of $R^2$ performance on the held-out test set of VAD dimensions.

valence–arousal–dominance (VAD) and emotion intensity dimensions as auxiliary signals. We referred to this setting as Human-LLM Alignment with VAD-Guided Representations.

### 5.2.1 EMOTION AWARE RETRIEVAL MODEL

We built an **Emotion-Aware Retrieval Model** (Figure 5), where the *primary task* is bidirectional retrieval between brain activity and text representations, optimized with a contrastive objective. In addition, we incorporated emotion dimension prediction as an *auxiliary task* to provide affective supervision. We used Qwen2.5-72B-Instruct as the text encoder in our framework for its best performance in evaluation.

**Encoders.** Brain ROI $\beta$-values $X_b \in \mathbb{R}^{N \times D_b}$ and text representations $X_t \in \mathbb{R}^{N \times D_t}$ are projected by modality-specific MLPs:

$$H_b = \text{norm}(f_b(X_b)), \quad H_t = \text{norm}(f_t(X_t)), \tag{1}$$

where $\text{norm}(\cdot)$ denotes $\ell_2$ normalization. The similarity matrix is then computed as $S = H_b H_t^\top$.

**Multi-positive contrastive loss.** For sample $i$ with positive set $\mathcal{P}(i)$, the brain→text InfoNCE (Oord et al., 2018) objective is combined with its symmetric text→brain counterpart, yielding

$$\mathcal{L}_{\text{MP}}^{b \to t} = -\frac{1}{N} \sum_{i=1}^{N} \log \frac{\sum_{j \in \mathcal{P}(i)} \exp(S_{ij}/\tau)}{\sum_{j=1}^{N} \exp(S_{ij}/\tau)}, \quad \mathcal{L}_{\text{ctr}} = \tfrac{1}{2}\left(\mathcal{L}_{\text{MP}}^{b \to t} + \mathcal{L}_{\text{MP}}^{t \to b}\right). \tag{2}$$

**Emotion-aware auxiliary tasks.** Emotion supervision includes VAD (valence, arousal, dominance) and intensity prediction. For both tasks, regression heads are applied to the separate embeddings of brain and text, and trained using mean squared error (MSE) against the annotated targets.

**Overall objective.** The training objective is a weighted combination:

$$\mathcal{L} = \lambda_{\text{ctr}}\mathcal{L}_{\text{ctr}} + \lambda_{\text{VAD}}\mathcal{L}_{\text{VAD}} + \lambda_{\text{Int}}\mathcal{L}_{\text{Int}}. \tag{3}$$

**Variants.** We evaluated several variants of the proposed model: (i) **Basic**, trained only with contrastive loss; (ii) **Pred_Only**, where emotion features are used solely for auxiliary prediction; (iii) **Text_Only** and **Brain_Only**, where emotion features are injected into text or brain embeddings, respectively; (iv) **Pred_Only_Drop**, a regularized version of Pred_Only with dropout applied to emotion features. The pseudocode for model training can be found in the Appendix E.1.

### 5.2.2 MODEL TRAINING AND RESULTS

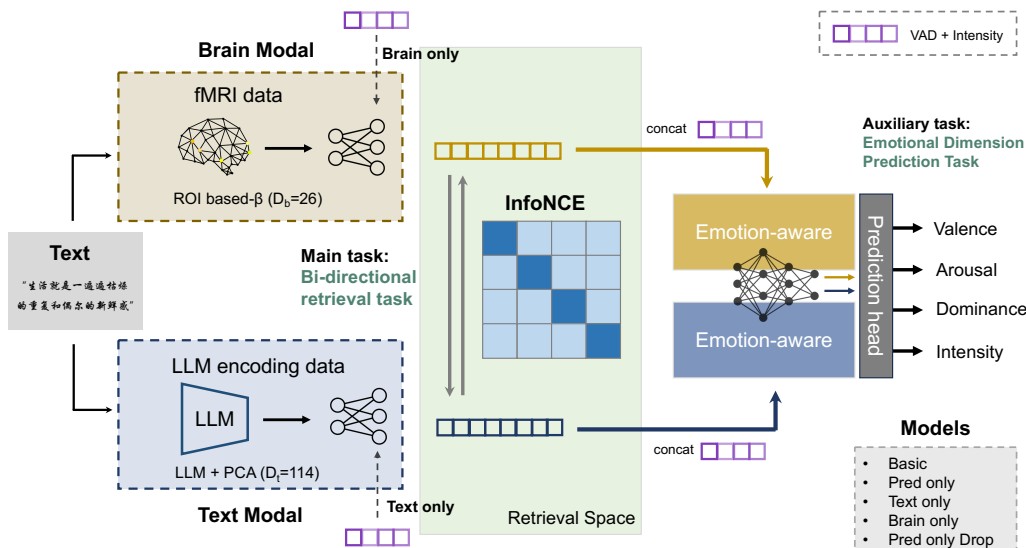

Figure 5: Emotion-Aware Retrieval Model.

Table 4 summarizes the bidirectional retrieval results. The chance level corresponds to random retrieval given the number of candidates ($1/24 = 0.042$ for Hit@1 and $3/24 = 0.125$ for Hit@3 in our setting).

**Baseline.** The Basic model without emotion supervision achieves limited performance (Hit@1 = 0.082/0.090), only slightly above the chance level.

Table 4: Retrieval performance.

| Model | Hit@1 | | Hit@3 | |
|---|---|---|---|---|
| | B→T | T→B | B→T | T→B |
| Basic | 0.082 | 0.090 | 0.203 | 0.262 |
| Pred_Only | 0.098 | 0.096 | 0.240 | 0.258 |
| Text_Only | 0.152 | 0.092 | 0.345 | 0.279 |
| Brain_Only | **0.169** | **0.167** | **0.375** | **0.414** |
| Pred_Only_Drop | 0.120 | 0.108 | 0.259 | 0.285 |

Note: B = Brain and T = Text.

**Prediction-only supervision.** Simply adding auxiliary VAD and intensity prediction (Pred_Only) does not enhance retrieval (Hit@1 = 0.098/0.096), suggesting that prediction heads alone provide weak guidance for cross-modal alignment.

**Emotion injection.** Incorporating emotional features into text (Text_Only) or brain (Brain_Only) embeddings substantially improves retrieval. In particular, the Brain_Only variant achieves the best results (Hit@1 = 0.169/0.167, Hit@3 = 0.375/0.414), highlighting the strong utility of affective cues for neural representations.

**Robustness and Statistical Evidence.** To ensure stability, we repeated all alignment experiments using multiple random seeds (1, 3, 5, 7, 42, 100). We further conducted formal statistical tests on Hit@1 and Hit@3 across all evaluated models. An ANOVA revealed a significant overall model effect ($F_{(4,25)} = 6.622$, $p < 0.001$). Post-hoc analyses confirmed that the BRAIN_ONLY model not only achieved the highest retrieval accuracy but also significantly outperformed the Basic model in both Hit@1 ($p_{\text{adj}} = 0.017$) and Hit@3 ($p_{\text{adj}} = 0.001$).

**Impact on Learned Representations.** We analyzed the learned embeddings in terms of sensitivity, disentanglement, and entropy. As shown in Figure 6, emotion-aware models consistently surpass the Basic variant across all metrics, indicating that affective supervision induces a stronger alignment of brain and language embeddings within a shared emotional space. Further interpretability analyses are provided in Appendix E.2.

## 6 DISCUSSION

Our dataset provides both academic and practical value. It enables future research on emotion representation in language and the brain, and can be directly applied to the training of large language models and embodied agents with emotion-related capabilities, particularly for recognizing and understanding nuanced emotions in Chinese. Beyond categorical annotations, SinoMultiAf-

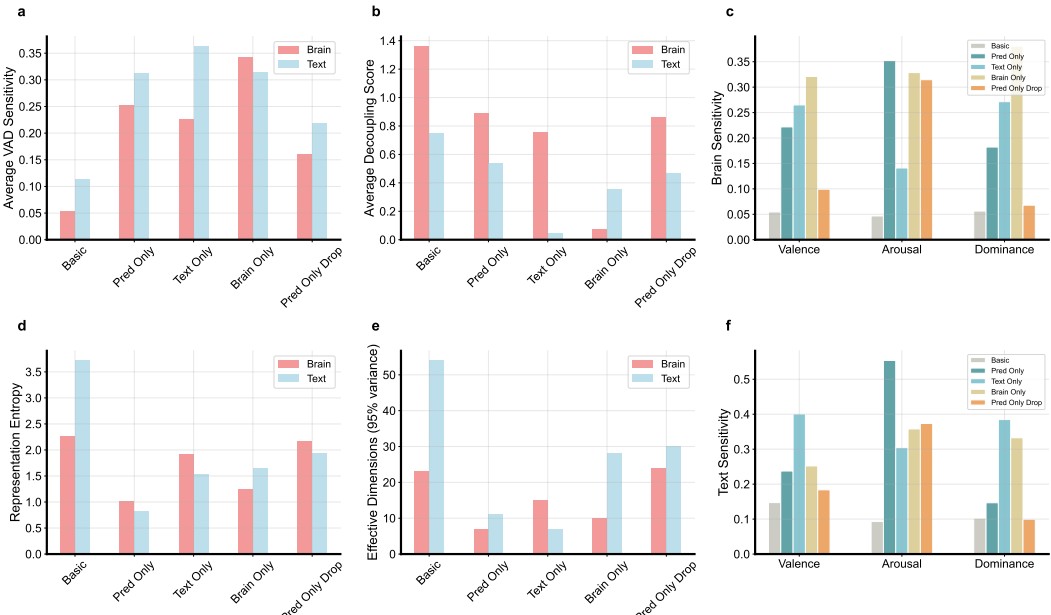

Figure 6: **Representation analysis.** Emotion-aware models show higher sensitivity, lower disentanglement (reflecting integration of emotion), and reduced entropy (suggesting more structured embeddings) compared to the Basic model.

fect includes emotion intensity scores and continuous valence–arousal–dominance (VAD) ratings, thereby supporting both discrete and dimensional perspectives of affect. The presence of semantically related yet distinct emotion categories is typical in fine-grained emotion taxonomies and further enriches the understanding of subtle affective differences. Psychological studies show that the ability to use and discriminate a rich set of emotion words, known as emotional granularity, is associated with better emotion regulation and mental well-being (Kashdan et al., 2015). From this perspective, our dual-level design not only preserves theoretically meaningful fine-grained distinctions but also provides practically valuable supervision signals that enable models to capture the subtleties of emotional expression in language. We anticipate that SinoMultiAffect will foster both theoretical advances and practical applications in emotion-centered artificial intelligence.

Finally, although fMRI is not yet practical for large-scale deployment, prior studies demonstrate that incorporating neural signals during fine-tuning can improve a model's alignment with human semantic and conceptual representations, as well as its generalization and interpretability (Schwartz et al., 2019; Moussa et al., 2024; Negi et al., 2025; Muttenthaler et al., 2025). In the emotion domain, the human brain exhibits embodied and spontaneous responses that cannot be fully captured by text labels alone (Baldassano et al., 2017; Barrett, 2017b). Thus, fMRI data collected during emotional text processing can serve as a grounded and biologically meaningful supervision signal, helping future models better capture the nuances of human emotional experience. Our overarching goal is to preserve both the consensus and natural variability of fine-grained emotional expressions, enabling LLMs to more effectively understand, differentiate, and support human emotions in real-world interactions.

## 7 CONCLUSION

We introduced **SinoMultiAffect**, a Chinese multi-modal dataset combining fine-grained emotion labels with fMRI recordings, enabling alignment of language and neural emotion representations. We reported zero-shot results of 13 open-source LLMs and proposed a VAD-guided cross-modal framework, showing that affective supervision strongly aligns brain and language embeddings. The dataset provides a foundation for culturally grounded affective computing and advances research on human–AI alignment in emotion understanding.

ETHICS STATEMENT

This work adheres to the ICLR Code of Ethics. Texts in our dataset were collected from publicly available social media platforms. To ensure that no private or sensitive information was disclosed, we applied an automatic masking procedure using a large language model (LLM), followed by manual verification by two researchers. The fMRI experiment was conducted under the approval of the hospital ethics committee, and all participants provided informed consent prior to scanning. Data were anonymized and stored securely in accordance with conference ethical guidelines.

REPRODUCIBILITY STATEMENT

We have taken several steps to ensure the reproducibility of our work.

(1) **Datasets.** All datasets used in this paper are either publicly available or will be released upon publication. The newly constructed **SinoMultiAffect** dataset, together with the original annotation results and detailed labeling protocols, will be made publicly accessible after publication. During the review phase, the dataset is provided in the supplementary material to facilitate verification.

(2) **Code and Implementation.** We have released our complete codebase in an anonymous repository for review[3]. The repository contains data preprocessing, model training, and evaluation scripts, as well as a requirements file specifying the software environment. Upon publication, the repository will be made fully public.

(3) **Model and Training Details.** The architecture of our models and all training configurations (e.g., embedding dimension, temperature, loss weights, optimizer, batch size, learning rate schedule, and number of epochs) are explicitly specified in the configuration file included in the repository.

(4) **Evaluation Protocol.** We reported results averaged across multiple random seeds to reduce variance, and provided explicit formulas for evaluation metrics in the appendix to ensure clarity and replicability.

LLM USAGE STATEMENT

We acknowledge the use of large language models (LLMs) to assist in the preparation of this paper. Specifically, LLMs were employed to polish the language, improve clarity and readability, and correct grammatical or stylistic errors in the paper. All substantive ideas, experimental designs, analyses, and conclusions presented in this work are the original contributions of the authors.

---

[3]https://anonymous.4open.science/r/SinoMultiAffect_anonymous-2649

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

# APPENDIX

## A   EMOTION TAXONOMY

### A.1   EMOTION TAXONOMY AND DEFINITION

Emotion categories used in our dataset and their brief meaning presented as instructions for raters are shown in Table S1 (English translation version).

Table S1: Emotion categories and their brief meanings.

| Category | Brief Meaning |
|---|---|
| **Positive Emotions** | |
| joy | Pleasure; happiness. |
| love | Affection; deep emotional attachment to people or things. |
| looking forward | Holding positive expectations for the future. |
| pride in oneself | Feeling very pleased or satisfied with oneself. |
| optimism | Positive outlook toward the future; full of hope. |
| appreciation | Acknowledgment and enjoyment of the value of a person or work. |
| satisfaction | A sense of peace after achieving goals or meeting needs. |
| trust | A sense of safety and reliability in others. |
| sympathy | Concern and understanding for others' difficulties. |
| being at ease | A relaxed state both physically and mentally. |
| healing | A warm feeling of emotional comfort and recovery. |
| gratitude | Deep thankfulness for others' help, care, or kindness. |
| being moved | Deep resonance and warmth from others' kindness or sincerity. |
| admiration | Desire for others' advantages, achievements, or happiness. |
| **Negative Emotions** | |
| anger | Intense emotion caused by injustice or offense. |
| sadness | Emotion caused by loss, failure, or pain. |
| disgust | Strong aversion toward someone or something. |
| fear | Feeling of being scared in the face of threat or danger. |
| disappointment | Negative feeling after expectations are not met. |
| remorse | Feeling regretful about past actions. |
| pity | Sense of loss or sorrow for unfulfilled wishes or missed chances. |
| guilt | Self-blame for hurting others or violating morals. |
| envy | Unease when others possess something one lacks. |
| anxiety | Nervousness and unease about uncertainty in the future. |
| embarrassment | Psychological discomfort in social or awkward situations. |
| shyness | Feeling nervous or uneasy in social situations. |
| cynicism | Lack of trust or hope toward society, others, or the world. |
| pessimism | Negative expectations for the future. |
| **Other Emotions** | |
| surprise | Reaction to unexpected events. |
| curiosity | Desire to know or explore new things. |
| alarm | Heightened attention to potential risks or abnormalities. |
| confusion | Not understanding complex or unclear information. |
| hesitant | Difficulty in making a choice. |
| nostalgia | Emotional recollection of good times in the past. |
| neutral | Typically an informative or objective expression without strong emotions. |

## A.2 MAPPING FROM 35 FINE-GRAINED EMOTIONS TO 17 COARSE CATEGORIES

The hierarchical mapping from the 35 fine-grained emotion labels to the 17 coarse-grained categories is provided in Table S2 (English translation version). This hierarchical labeling scheme enhances usability without compromising granularity. Under the new coarse taxonomy, the Jaccard index increases to 0.486 across all 4,058 samples.

| English Superclass | English Subclass |
|---|---|
| Joy | Joy |
| Joy | Self-satisfaction |
| Joy | Optimism |
| Joy | Satisfaction |
| Joy | Being at ease |
| Joy | Healing |
| Affection | Love |
| Affection | Appreciation |
| Affection | Being Moved |
| Affection | Admiration |
| Affection | Sympathy |
| Praise | Gratitude |
| Trust | Trust |
| Wishing | Looking Forward |
| Anger | Anger |
| Anger | Cynicism |
| Sadness | Sadness |
| Sadness | Pessimism |
| Disappointment | Disappointment |
| Disappointment | Pity |
| Guilt | Remorse |
| Guilt | Guilt |
| Longing | Nostalgia |
| Fear | Fear |
| Fear | Alertness |
| Shyness | Shyness |
| Annoyance | Anxiety |
| Annoyance | Embarrassment |
| Annoyance | Confusion |
| Disgust | Disgust |
| Criticism | Hesitant |
| Envy | Envy |
| Surprise | Surprise |
| Surprise | Curiosity |

Table S2: Mapping between English Emotion Superclasses and Subclasses.

# B  INTERRATER CONSISTENCY

We calculated **interrater correlation** in each category as follows: for each emotion, we first construct a sentence–annotator matrix that records whether each annotator selected the category for each sentence. Then, for every annotator, we calculate the Spearman rank correlation between their own labels and the mean labels of the remaining annotators on the same sentences. The interrater correlation of a category is defined as the average of these correlations across all annotators.

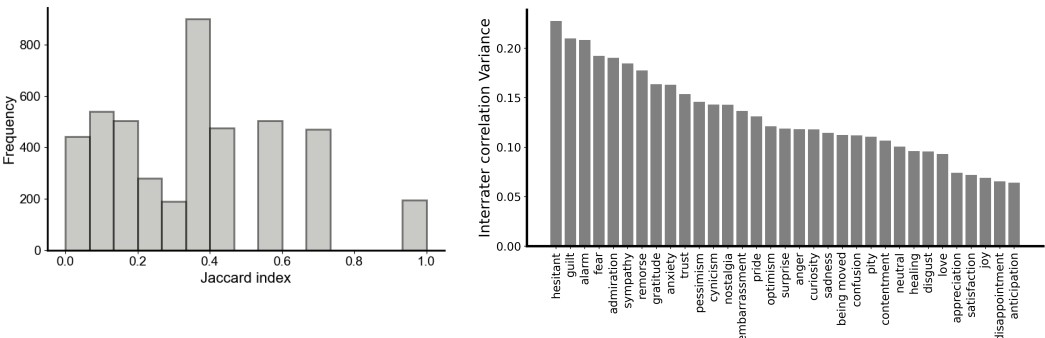

Figure S1: Interrater Agreement Analysis.

The distribution of Jaccard index scores across all sentences and interrater correlation variance in each emotion are shown in Figure S1. *Appreciation* and *disappointment* have the most consistent labeling, whereas *hesitant*, *guilt*, and *alarm* demonstrate the largest interrater variability.

# C  FMRI SCANNING AND DATA PREPROCESSING

Stimuli were presented for 8 s followed by a jittered inter-trial interval (13-17s for each trial in total). Each participant completed three runs of 40 trials, resulting in 120 sentences in total. fMRI data were acquired on a 3T Siemens Prisma scanner (TR = 2 s, TE = 30 ms, flip angle = 90°, voxel size = 3 × 3 × 3 mm³, 50 slices). High-resolution T1-weighted anatomical images were also collected. Preprocessing was performed using SPM12, including slice timing correction, realignment, normalization to MNI space, and 6 mm FWHM smoothing. Data were organized following the BIDS standard. The full BIDS-formatted fMRI dataset will be released upon publication, while the embedding features used for model training are already available in the **Supplementary File**. We provide both ROI-averaged beta values (covering emotion- and language-related regions defined by the Harvard–Oxford cortical and subcortical structural atlases; see Figure S2 for the full list of ROIs) and voxel-wise beta values for each stimulus and subject.

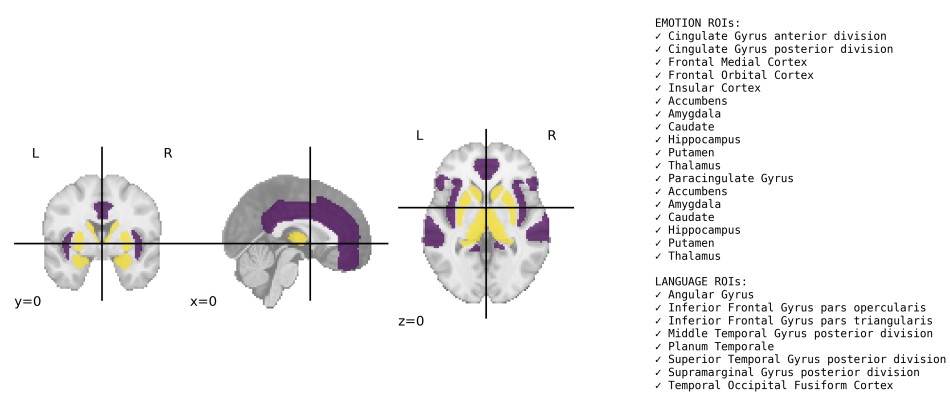

Figure S2: ROIs selected for neural representation.

# D EVALUATING LLMS ON NUANCED EMOTION RECOGNITION

## D.1 PROMPTS

We adopt the architecture of the *Choice Funnel* prompt, as proposed by Xu et al. (2025), but customize its content for the current study. The core structural elements of the original prompt are preserved, while the specific queries, task definitions, and constraints are rigidly defined and localized to address our multi-label emotion classification task. Critically, the language of the final prompt has been comprehensively converted to Chinese. The resulting prompt template used throughout our experiments is shown below:

---

任务：请分析以下语句，并判断其所反映的情绪。

语句：
{sentence}

可用情绪选项：
A.喜悦
B.爱
......

要求：请从"可用情绪选项"中选择所有语句中反映的情绪。

答案：

---

The English translation of the prompt template is presented below:

---

Task: Please analyze the following sentence and determine the emotions reflected in it.

Sentence:
{sentence}

Available Emotion Options:
A. Joy
B. Love
...

Requirement: Please select all emotions reflected in the sentence from the "Available Emotion Options".

Answer:

---

## D.2 METRICS

We evaluate model performance using several standard metrics for multi-label recognition. For a given sample $i$, let $Y_i$ be the set of true labels and $\hat{Y}_i$ be the set of predicted labels. For a specific class

$c$, let $TP_c$, $FP_c$, and $FN_c$ denote the counts of true positives, false positives, and false negatives, respectively.

The first metric, **Jaccard Similarity**, measures the intersection over the union of the predicted and true label sets for each sample, averaged over all $N$ samples:

$$\text{Jaccard} = \frac{1}{N} \sum_{i=1}^{N} \frac{|Y_i \cap \hat{Y}_i|}{|Y_i \cup \hat{Y}_i|}$$

Next, we report the **Precision** and **Recall**. To ensure all classes contribute equally regardless of their frequency, we compute these metrics by averaging the scores across all classes. First, we calculate the precision ($P_c$) and recall ($R_c$) for each class $c$ individually:

$$P_c = \frac{TP_c}{TP_c + FP_c} \quad , \quad R_c = \frac{TP_c}{TP_c + FN_c}$$

The final Precision and Recall are then obtained by taking the unweighted average of these per-class scores over all $C$ classes:

$$\text{Precision} = \frac{1}{C} \sum_{c=1}^{C} P_c \quad , \quad \text{Recall} = \frac{1}{C} \sum_{c=1}^{C} R_c$$

We also employ the **Exact Match (EM)** ratio, which is the strictest metric. It calculates the percentage of samples where the predicted set of labels exactly matches the ground truth set:

$$\text{EM} = \frac{1}{N} \sum_{i=1}^{N} \mathbb{I}(Y_i = \hat{Y}_i)$$

where $\mathbb{I}(\cdot)$ is the indicator function.

Moreover, to assess how closely model-derived emotion representations mirror human affective structure, we compute the Spearman rank correlation between the human similarity matrix $H$ and the model-derived similarity matrix $M$:

$$\text{Spearman}(H, M) = \rho_{\text{rank}}(H, M).$$

To quantify how well fine-grained emotion categories form separable clusters, we compute the average silhouette score over all samples:

$$s(i) = \frac{b(i) - a(i)}{\max\{a(i), b(i)\}}, \qquad \text{Silhouette} = \frac{1}{N} \sum_{i=1}^{N} s(i).$$

To examine whether model representations encode continuous affective dimensions, we evaluate the $R^2$ of predicting valence, arousal, and dominance on a held-out test set:

$$R_d^2 = 1 - \frac{\sum_i (y_{d,i} - \hat{y}_{d,i})^2}{\sum_i (y_{d,i} - \bar{y}_d)^2}, \qquad d \in \{\text{V}, \text{A}, \text{D}\}.$$

## D.3 EXTENDED PERFORMANCE ANALYSIS

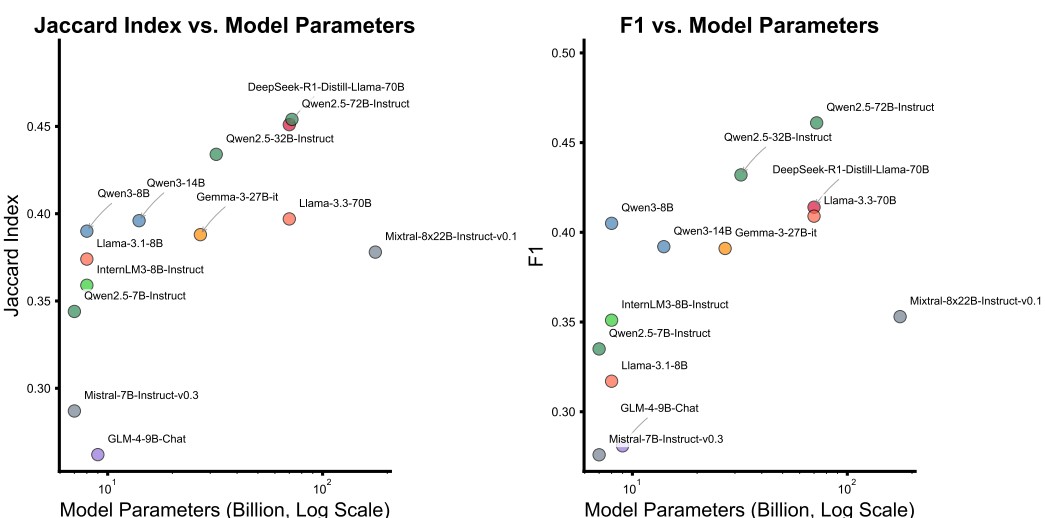

Figure S3: Key Performance Metrics versus Model Parameter Count on a Logarithmic Scale.

As shown in Figure S3, models with larger parameter counts generally achieve higher Jaccard and F1 scores, while the Mixture-of-Experts model Mixtral-8x22B performs comparably to smaller dense models. This may reflect limitations in its training data's linguistic diversity (e.g. relatively lower representation of Chinese or other non-European languages), which might constrain its performance on our Chinese-centric dataset.

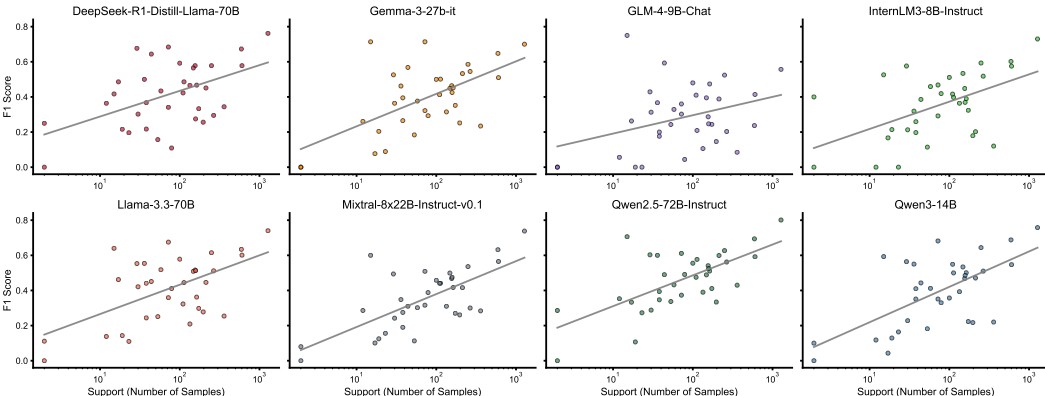

Figure S4: Per-emotion F1 score versus label support (number of samples) for each evaluated model.

Besides, our analysis shows that model performance tends to improve with the number of available samples per emotion category Figure S4, highlighting the challenge of recognizing low-frequency, fine-grained emotions.

Building on the observation that model performance scales with the frequency of emotion categories, Figure S5 reveals how models perform on each specific emotion category. Certain common and semantically distinct emotions, such as *joy*, *satisfaction*, *disappointment*, and *appreciation*, are recognized with relatively high F1 scores across most models. In contrast, performance is consistently poor for emotions that are more nuanced, abstract, or infrequent, including *cynicism*, *remorse*, *pity*, and *trust*, where F1 scores often approach zero. High-performing categories typically correspond to foundational emotions that are directly expressed through conventional language, for which pre-trained models possess robust representations. Conversely, low-performing emotions are often socially complex and conveyed through subtle, indirect, or ironic expressions, posing greater challenges for generalization.

Finally, we found that all evaluated models produce more emotion labels per sample than the consensus ground-truth average (1.393), and some of these cases may reflect genuine ambiguity in emotional interpretation rather than discrepancies between model and human judgments. To better understand this phenomenon, we conducted an additional analysis that compares LLM predictions with all human-provided labels, rather than only the consensus set (see table S3). This analysis captures the full range of human emotional interpretations, allowing us to examine how model outputs relate to the variability observed among annotators in ambiguous cases.

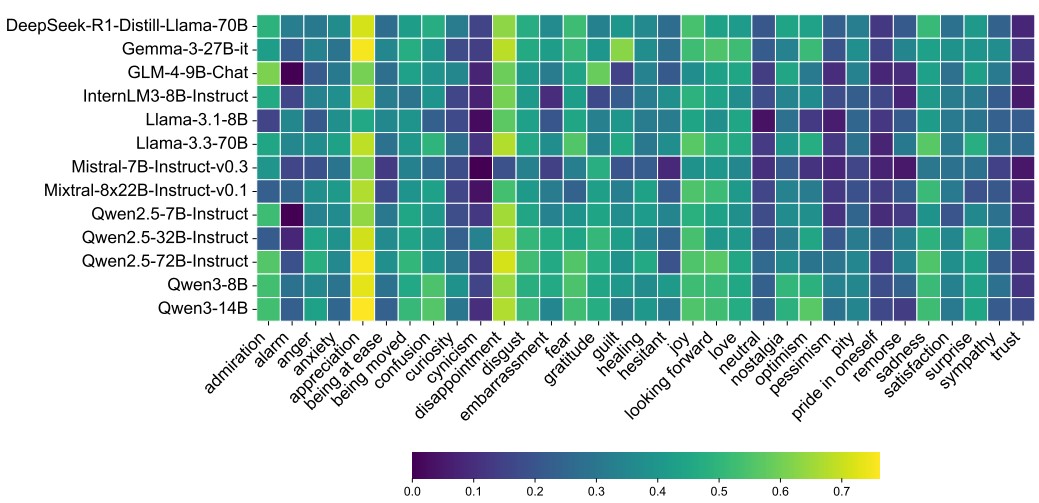

Figure S5: Emotion F1 Scores Heatmap.

Table S3: Comparison of model performance using all human-provided labels versus consensus labels across multiple evaluation metrics.

| Model Name | Jaccard_all | Jaccard_cons. | Prec_all | Prec_cons. | Rec_all | Rec_cons. | EM_all | EM_cons. |
|---|---|---|---|---|---|---|---|---|
| Qwen2.5-72B-Instruct | **0.409** | **0.454** | **0.640** | **0.413** | **0.329** | **0.583** | **0.066** | **0.163** |
| Qwen2.5-7B-Instruct | 0.279 | 0.344 | 0.552 | 0.368 | 0.246 | 0.410 | 0.024 | 0.149 |
| Qwen3-8B | 0.351 | 0.390 | 0.607 | 0.398 | 0.306 | 0.533 | 0.045 | 0.125 |
| Qwen3-14B | 0.352 | 0.396 | 0.578 | 0.374 | 0.269 | 0.507 | 0.044 | 0.123 |
| Llama-3.3-70B-Instruct | 0.367 | 0.397 | 0.590 | 0.385 | 0.323 | 0.547 | 0.048 | 0.108 |
| gemma-3-27b-it | 0.388 | 0.388 | 0.558 | 0.350 | 0.325 | 0.509 | 0.054 | 0.105 |
| internlm3-8b-instruct | 0.333 | 0.359 | 0.541 | 0.365 | 0.283 | 0.458 | 0.038 | 0.090 |
| GLM-4-9B-Chat | 0.282 | 0.262 | 0.516 | 0.366 | 0.250 | 0.369 | 0.028 | 0.051 |

# E  HUMAN-LLM ALIGNMENT

## E.1  ALGORITHMIC DETAILS OF MODEL TRAINING AND EVALUATION

**Overview.** This appendix details the training and evaluation procedures for the Emotion-Aware Cross-Modal Retrieval framework. We denote brain and text inputs as $\mathbf{b}_i$ and $\mathbf{t}_i$, their encoders as $f_B(\cdot)$ and $f_T(\cdot)$, and the emotion dimension targets as $\mathbf{v}_i \in \mathbb{R}^3$ (valence, arousal, dominance) with optional intensity $s_i$. Cosine similarity is computed on $\ell_2$-normalized embeddings.

**Training algorithm.** Given batches of paired samples, we obtain modality-specific representations $\mathbf{h}_i^B = f_B(\mathbf{b}_i)$ and $\mathbf{h}_i^T = f_T(\mathbf{t}_i)$. If the attention module is enabled, we derive cross-modulated embeddings $\tilde{\mathbf{z}}_i^B, \tilde{\mathbf{z}}_i^T$; otherwise we use the original $\mathbf{h}_i^B, \mathbf{h}_i^T$. The retrieval objective is a *symmetrized multi-positive* InfoNCE over the similarity matrix $S_{ij} = \cos(\tilde{\mathbf{z}}_i^B, \tilde{\mathbf{z}}_j^T)$ with temperature $\tau$, where $\mathcal{P}(i)$ indexes the within-batch positives for item $i$. Two auxiliary heads regress VAD and (optionally) predict intensity on both modalities, yielding losses $L_{\text{vad}}$ and $L_{\text{int}}$. The total objective is a weighted sum $L = \lambda_{\text{retr}} L_{\text{retr}} + \lambda_{\text{vad}} L_{\text{vad}} + \lambda_{\text{int}} L_{\text{int}}$. We train with standard optimizers and a learning-rate scheduler, validating each epoch and selecting the checkpoint that maximizes Hit@1 on the validation set.

**Evaluation protocol.** For evaluation, we encode all samples once (no gradient) to obtain $\mathbf{z}_i^B$ and $\mathbf{z}_i^T$ and compute the full similarity matrix $S$. Retrieval is evaluated in both directions (Brain→Text and Text→Brain) by ranking candidates per query and reporting Hit@K for $K \in \{1, 3\}$. For emotion dimensions, we report the coefficient of determination $R^2$ for each of valence, arousal, and dominance, as well as their average. The same preprocessing and normalization used in training are applied at evaluation.

Table S4: Unified pseudo-code for training and evaluation of the Emotion-Aware Retrieval model.

| **Algorithm: Emotion-Aware Cross-Modal Retrieval (Training & Evaluation)** |
|---|
| **Input:** datasets $\{\mathcal{D}_{train}, \mathcal{D}_{val}, \mathcal{D}_{test}\}$, training hyperparameters 
 **Output:** trained parameters $\Theta^\star$, evaluation metrics (Hit@K, $R^2$) |
| **Training Procedure:** 
 1. Initialize model $\mathcal{M}_\Theta$, optimizer, and scheduler. 
 2. **For** each epoch $e = 1, \ldots, E$: 
   2.1 Sample mini-batch $\{(\mathbf{b}_i, \mathbf{t}_i, \mathbf{v}_i, s_i)\}_{i=1}^{B}$. 
   2.2 Encode features: $\mathbf{h}_i^B = f_B(\mathbf{b}_i), \quad \mathbf{h}_i^T = f_T(\mathbf{t}_i)$. 
   2.3 Apply cross-modal attention (if enabled): 
     $\tilde{\mathbf{z}}_i^B = \text{Attn}(\mathbf{h}_i^B, \mathbf{h}_i^T), \quad \tilde{\mathbf{z}}_i^T = \text{Attn}(\mathbf{h}_i^T, \mathbf{h}_i^B)$. 
     Otherwise, set $\tilde{\mathbf{z}}_i^B = \mathbf{h}_i^B, \tilde{\mathbf{z}}_i^T = \mathbf{h}_i^T$. 
   2.4 Predict auxiliary outputs: $\widehat{\mathbf{v}}_i^B, \widehat{\mathbf{v}}_i^T$ (VAD), $\hat{s}_i^B, \hat{s}_i^T$ (intensity). 
   2.5 Compute similarity matrix: $S_{ij} = \cos(\tilde{\mathbf{z}}_i^B, \tilde{\mathbf{z}}_j^T)$. 
   2.6 Retrieval loss (InfoNCE): $L_{\text{ctr}}$ 
   2.7 Auxiliary VAD regression: $L_{\text{VAD}}$ 
   2.8 Intensity prediction loss (optional): $L_{\text{Int}}$ 
   2.9 Total loss: $L = \lambda_{\text{ctr}} L_{\text{ctr}} + \lambda_{\text{VAD}} L_{\text{VAD}} + \lambda_{\text{Int}} L_{\text{Int}}$. 
   2.10 Update parameters $\Theta \leftarrow \Theta - \eta \nabla_\Theta L$. 
   2.11 Validate on $\mathcal{D}_{val}$ and save best checkpoint (based on Hit@1). 
 3. Return $\Theta^\star$. |
| **Evaluation Procedure:** 
 1. Encode all test samples: $\mathbf{z}_i^B = f_B(\mathbf{b}_i), \mathbf{z}_i^T = f_T(\mathbf{t}_i)$. 
 2. Compute similarity matrix $S_{ij} = \cos(\mathbf{z}_i^B, \mathbf{z}_j^T)$. 
 3. For each sample $i$: rank candidates and compute Hit@K (B→T, T→B). 
 4. Compute $R^2$ for valence, arousal, dominance, and intensity: 
   $$R_d^2 = 1 - \frac{\sum_i (\hat{v}_{i,d} - v_{i,d})^2}{\sum_i (v_{i,d} - \bar{v}_d)^2}, \ d \in \{\text{val}, \text{aro}, \text{dom}, \text{int}\}.$$ 
 5. Report per-dimension $R^2$ and averaged score. |

## E.2 INTERPRETABILITY ANALYSIS

As shown in Table S5, incorporating affective supervision (Brain_Only) markedly improves the alignment between embedding geometry and affective dimensions compared to the Basic variant. The **Arousal** dimension shows the most consistent gains across both brain ($\Delta = +0.3412$) and text embeddings ($\Delta = +0.2434$), indicating that the proposed model captures arousal-related gradients in the representational space more effectively. This pattern is also visually evident in Figure S6, where embeddings exhibit a clear continuous organization along the arousal axis. We quantify "Spatial–VAD correlation" as the Spearman rank correlation between all pairwise Euclidean distances in the 2D PCA embedding and the corresponding absolute pairwise differences of a given V/A/D value:

$$\rho = \text{corr}_{\text{Spearman}}\Big( D_X(i,j),\ D_v(i,j) \Big), \quad D_X(i,j) = \|\mathbf{x}_i - \mathbf{x}_j\|_2, \quad D_v(i,j) = |v_i - v_j|.$$

Higher $\rho$ indicates that items with similar VAD values lie closer in the embedding; *Delta* reports (*Brain_Only* − *Basic*).

| Dimension | Basic | Brain-only | Delta |
|---|---|---|---|
| *Brain Embedding Performance (Spatial-VAD Correlation)* | | | |
| Valence | 0.0228 | 0.2112 | +0.1884 |
| Arousal | 0.0076 | 0.3489 | **+0.3412** |
| Dominance | 0.0322 | 0.3183 | +0.2861 |
| *Text Embedding Performance (Spatial-VAD Correlation)* | | | |
| Valence | 0.0279 | 0.1440 | +0.1160 |
| Arousal | 0.0127 | 0.2562 | **+0.2434** |
| Dominance | 0.0602 | 0.2954 | +0.2352 |

Table S5: Spatial-VAD correlation for brain and text embeddings.

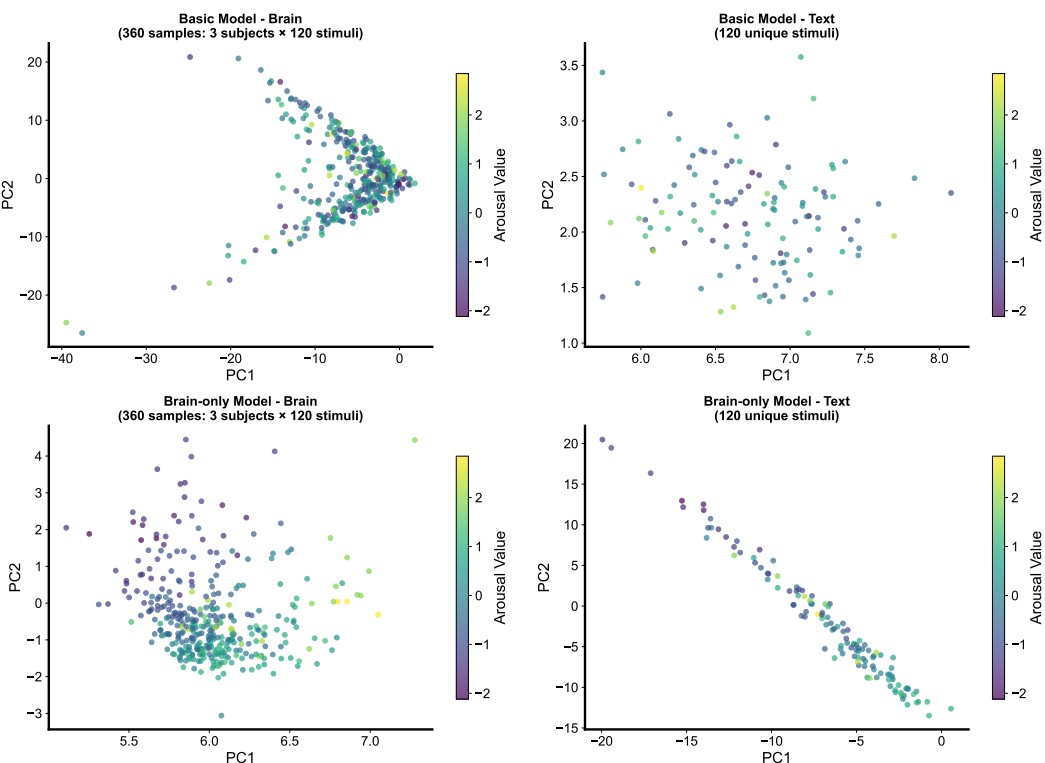

Figure S6: Arousal Dimension Visualization (PCA).

## F   LIMITATIONS

A current limitation of our work lies in the dataset scale, as the present release contains 4,500 samples. However, we have already established a mature annotation platform and a standardized pipeline for data processing and analysis, which will support continuous expansion of the resource. Future versions will include larger volumes of annotated texts and fMRI recordings, and we plan to release both the raw annotations and consensus-labeled sentences in a database format on our website, enabling researchers to directly access up-to-date versions of the dataset. This ongoing effort will further enhance the robustness, generalization, and usability of SinoMultiAffect, providing a stronger foundation for both theoretical exploration and practical development of emotion-related AI.

