# OpenReview forum: "SinoMultiAffect: A Chinese Multi-Label and Fine-Grained Emotional Text Dataset with fMRI Data"
_ICLR.cc/2026/Conference — Submitted to ICLR 2026_

### Official Review · Reviewer_92LJ · 2025-10-25

**Soundness:** 2
**Presentation:** 3
**Contribution:** 2
**Rating:** 2
**Confidence:** 5

**Summary:**

The paper presents SinoMultiAffect, a Chinese multimodal emotion dataset that combines text and functional magnetic resonance imaging (fMRI) data. The dataset is used to benchmark large language models in multi-label emotion recognition. The paper also proposes a method for aligning linguistic and neural representations of emotions using VAD (valence, arousal, dominance) features. This method contributes a new resource and framework for studying the correspondence between human and AI emotions.

**Strengths:**

1. The paper introduces SinoMultiAffect, a new Chinese multilabel and fine-grained emotional text dataset that expands upon prior affective resources for the language. instance is annotated with 35 categorical emotions as well as four continuous affective dimensions (Valence, Arousal, Dominance, and Intensity) enabling both discrete and dimensional analysis of emotion.
2. The collection of fMRI recordings for a subset of sentences adds a unique neuroimaging modality, bridging linguistic and neural representations of affect.
3. The paper formulates a novel VAD-guided Human–LLM Alignment framework, exploring the alignment between large language model embeddings and human brain activity.
4. The authors provide code and data resources.

**Weaknesses:**

1. The proposed dataset is relatively small when compared to existing Chinese emotion corpora such as CH-MEAD, EmotionTalk, and ChineseEmoBank, which limits its representativeness and robustness.

2. Although the corpus is described as multimodal, it only includes two modalities - text and fMRI, without audio or visual data, which weakens the claim of true multimodality.
3. While comparing multiple large language models for zero-shot emotion recognition may be topical, it does not provide significantly new insights, as this has been widely explored in previous studies.
4. The use of large LLM embeddings for alignment may not be fully justified, as simpler BERT-based emotion encoders could be more efficient feature extractors.
5. The paper lacks explicit information about the prompts used for each LLM and does not compare with previous prompt designs, limiting the reproducibility of the study.
6. The evaluation of LLMs is limited to categorical emotion recognition, whereas the proposed VAD-guided Human–LLM Alignment framework applies only to dimensional (valence–arousal–dominance–intensity) emotion modeling. This lack of cross-validation across these two areas reduces the conceptual consistency of the experimental design.
7. There is no comparison with existing emotion recognition or brain–text alignment methods, making it challenging to assess the practical significance of the proposed approach.

**Questions:**

1. Could the authors please provide more information about the specific prompt templates used for each language model in the zero-shot emotion recognition task?
2. Why was it decided to evaluate LLMs only on categorical emotions, rather than on the valence, arousal, dominance, and intensity dimensions, or the VAD-guided human-LLM alignment framework for valence, arousal, and dominance recognition? What were the reasons for this decision?
3. Would it be possible for the authors to compare their alignment framework with simpler baselines using emotional BERT or RoBERTa embeddings, to demonstrate the necessity of LLM representations for achieving affective alignment?
4. Could the authors elaborate on the practical implications of the fMRI-based alignment framework, given its high cost and limited scalability for real-world applications?

---

> ### Author Response · Authors · 2025-11-21
> **General response**
>
> We sincerely thank Reviewer 92LJ for the detailed and constructive feedback.
>
> In the following response, we have provided responses to the major points including the dataset scale, the evaluation of fine-grained emotion recognition (Response to Weaknesses 1-3), the reporting of prompt templates (Response to Weakness 5 and Question 1), elaboration of practical implications of fMRI-based alignment (Response to Question 4) and the responses related to evaluation settings and alignment (Response to Weakness 6 and Question 2). Additional comments raised by the reviewer are addressed in their corresponding individual responses. We hope that our clarifications and analyses are able to address the reviewer’s concerns.

---

> > ### Author Response · Authors · 2025-11-21
> > **Response to Weaknesses 1-3:**
> >
> > **Weakness 1 (about small scale):**
> >
> > Thank you for pointing this out. We acknowledge that the current scale of our dataset (4,500 posts) is smaller than several large Chinese emotion corpora such as CH-MEAD, EmotionTalk. However, it is important to note that the design goals and annotation strategies of these datasets differ substantially from ours. First, the largest dataset to date, CMACD, was machine labeled, containing only 6 coarse emotion categories. Similarly, EmotionTalk also only includes 7 coarse emotion categories and assigns a single label per sentence. Both datasets therefore do not match the fine-grained, multi-label emotional distinctions that we aim to capture.
> >
> > It seems that the dataset most comparable to ours is CH-MEAD, which contains over 20,000 samples and includes 26 emotion categories. However, its stimuli span video, audio, and text modalities, meaning that emotion recognition can rely on additional sensory cues beyond language alone. In contrast, our goal is to isolate emotion signals expressed purely through text, as this is the modality through which large language models are most commonly accessed and used today. Therefore, CH-MEAD serves a different purpose and does not fully address the fine-grained textual emotion understanding that our dataset is designed to support.
> >
> > We are also considering expanding our dataset to increase the number of samples for different nuanced emotions, as our rating procedure has already become a standardized pipeline. We believe that this dataset will be valuable for researchers aiming to improve the emotion-related capabilities of LLMs that are more practical in everyday life.
> >
> > **Weakness 2 (about the statement of multimodal):**
> >
> > Thank you for raising this point. Our use of the term multimodal follows the convention adopted in prior work such as CLIP (Radford et al., 2021), where combining two different signals (e.g., text and images) is also referred to as a multimodal setting. In the same spirit, our corpus integrates text and fMRI, which we view as two distinct modalities.
> >
> > Reference:
> > 1. Radford, A., et al., (2021, July). Learning transferable visual models from natural language supervision. In International conference on machine learning (pp. 8748-8763). PmLR.
> >
> >
> > **Weakness 3 (about the new insight of zero-shot evaluation):**
> >
> > Thank you for your comment. While general emotion understanding in LLMs has been explored, their capability to recognize fine-grained emotions remains underexamined. Prior studies often focus on basic or coarse categories, whereas our work evaluates LLMs across 35 nuanced emotions.
> > Recognizing such distinctions is crucial for emotion regulation in humans (Gross, 2015, Kashdan et al., 2015), and developing this ability in LLMs could support more emotionally aware human-AI interactions. We believe our study provides new insights by addressing this finer level of emotional granularity.
> >
> > References:
> > 1. Kashdan, T. B., Barrett, L. F., & McKnight, P. E. (2015). Unpacking emotion differentiation: Transforming unpleasant experience by perceiving distinctions in negativity. Current Directions in Psychological Science, 24(1), 10-16.
> > 2. Gross, J. J. (2015). Emotion regulation: Current status and future prospects. Psychological inquiry, 26(1), 1-26.

---

> > ### Author Response · Authors · 2025-11-21
> > **Response to Weakness 7 (about lacking comparison with existing alignment methods):**
> >
> > Thank you for raising this point. The primary contribution of our work is the introduction of the SinoMultiAffect dataset, and the brain-text alignment analysis is intended as a proof-of-concept rather than a fully developed alignment method. Given that the alignment component is exploratory and not proposed as a new algorithmic framework, we did not position it as a method competing with existing brain-text alignment approaches.

---

> > ### Author Response · Authors · 2025-11-21
> > **Response to Question 4 (elaboration of practical implications of fMRI-based alignment):**
> >
> > We appreciate your concern regarding the high cost and limited scalability of fMRI data in real-world applications. While we agree that fMRI is not yet a practical modality for large-scale deployment, prior studies have shown that fine-tuning language models with brain signals can improve their alignment with human semantic representations (Schwartz et al., 2019; Moussa et al., 2025; Negi et al., 2025). Notably, recently a paper published in Nature has further proved this point, demonstrating that fine-tuning models using human neural signals can significantly improve alignment with human conceptual representations and boost generalization, robustness, and interpretability across tasks (Muttenthaler et al., 2025).
> >
> > In the context of emotion understanding, the human brain exhibits embodied and spontaneous responses that are often difficult to capture through text labels alone (Baldassano et al., 2017; Barrett, 2017). We therefore propose that fMRI data collected during emotional text processing can serve as a more grounded and biologically valid supervision signal, helping models better capture the nuances of human emotional experience in the future study.
> >
> > Reference:
> > 1. Moussa, O., Klakow, D., & Toneva, M. (2025). Improving semantic understanding in speech language models via brain-tuning. In Proceedings of the Thirteenth International Conference on Learning Representations (ICLR).
> > 2. Schwartz, D., Toneva, M., & Wehbe, L. (2019). Inducing brain-relevant bias in natural language processing models. Advances in neural information processing systems, 32. (NeurIPS)
> > 3. Negi, A., Oota, S. R., Nunez-Elizalde, A. O., Gupta, M., & Deniz, F. (2025). Brain-informed fine-tuning for improved multilingual understanding in language models. In Proceedings of the Thirty-ninth Annual Conference on Neural Information Processing Systems (NeurIPS).
> > 4. Muttenthaler, L., Greff, K., Born, F., Spitzer, B., Kornblith, S., Mozer, M. C., Müller, K.-R., Unterthiner, T., & Lampinen, A. K. (2025). Aligning machine and human visual representations across abstraction levels. Nature, 647(8089), 349–355.
> > 5. Baldassano, C., Chen, J., Zadbood, A., Pillow, J. W., Hasson, U., & Norman, K. A. (2017). Discovering event structure in continuous narrative perception and memory. Neuron, 95(3), 709-721. e705.
> > 6. Barrett, L. F. (2017b). The theory of constructed emotion: an active inference account of interoception and categorization. Social cognitive and affective neuroscience, 12(1), 1-23.

---

> > ### Author Response · Authors · 2025-11-22
> > **Response to Weaknesses 6, 4 and Questions 2, 3:**
> >
> > **Response to Weakness 6 and Question 2 (about settings of evaluation and alignment):**
> >
> > We appreciate the reviewer’s thoughtful question. Following your recommendation, we have added an evaluation of LLMs on the valence, arousal, and dominance dimensions. Specifically, for each model, we first extracted the sentence-level embedding and then trained a ridge regression model to predict each VAD dimension. We report the R² performance on the held-out test set, and the results are shown in the following table. This result indicates that models performing better on the fine-grained multi-label emotion classification task tend to achieve higher accuracy in VAD dimension prediction. We hope this additional analysis could provide a more comprehensive understanding of the models’ performance beyond categorical labels.
> >
> > | Model Name            | Valence pred | Arousal pred | Dominance pred | VAD avg |
> > |-----------------------|--------------|--------------|----------------|---------|
> > | Qwen2.5-72B-Instruct  | **0.782**        | 0.447        | 0.204          | 0.478   |
> > | Qwen2.5-32B-Instruct  | 0.772        | **0.453**        | **0.220**          | **0.481**   |
> > | Qwen2.5-7B-Instruct   | 0.738        | 0.437        | 0.213          | 0.463   |
> > | Qwen3-8B              | 0.730        | 0.447        | 0.205          | 0.461   |
> > | Qwen3-14B             | 0.735        | 0.418        | 0.144          | 0.432   |
> > | Llama-3.1-8B-Instruct | 0.683        | 0.441        | 0.138          | 0.420   |
> > | Llama-3.3-70B-Instruct| 0.736        | 0.411        | 0.168          | 0.438   |
> > | internlm3-8b-instruct | 0.762        | 0.438        | 0.214          | 0.471   |
> > | GLM-4-9B-Chat         | 0.730        | 0.436        | 0.186          | 0.451   |
> >
> >
> > Regarding the VAD-guided human-LLM alignment framework, at the current stage, our available fMRI dataset contains a limited number of samples, and the text stimuli associated with each fine-grained emotion category are relatively sparse. Using categorical labels under such conditions would substantially increase variance and reduce the stability of the alignment results. In contrast, using continuous VAD dimensions yields more reliable and robust alignment estimates given the small-sample setting. For this reason, we adopted the dimensional approach for the fMRI-based alignment analysis in the present study.
> >
> > **Response to Weakness 4 and Question 3 (about the use of pre-trained LM):**
> >
> > Thank you for the suggestion. Our focus on large LLM embeddings is intentional: we aim to study the fine-grained emotional capabilities of modern generative LLMs, which are the models most relevant for future human-AI emotional interaction. Since LLMs function as conversational systems rather than pure encoders, understanding and improving their alignment with human affective representations is central to our goal. Therefore, we prioritize evaluating and aligning LLM-specific embeddings rather than encoder-only models.

---

> ### Author Response · Authors · 2025-11-21
> **Response to Weakness 5 and Question 1 (about explicit prompt information):**
>
> Thank you for raising this important point. We agree that clearly reporting prompt templates is essential for reproducibility. In the revised manuscript, we have added a dedicated section describing all prompts used in the zero-shot emotion recognition experiments.
>
> **Old Prompt (Used in the initial submission):**
>
> > We originally adopted a straightforward multi-label instruction commonly used in prior LLM emotion-classification work:
> 情绪种类包括：{emotion_labels}。在上面的情绪种类中，分析以下句子包含哪些情绪，你最多可以选择三种情绪。待分析的句子：{sentence}
>
> English Translation:
> > Emotion categories include: {emotion_labels}. Among these categories, please analyze which emotions are present in the following sentence. You may select up to three emotions.
> Sentence to analyze: {sentence}
> While simple, this prompt design can lead to variability in multi-label prediction due to the open-ended generation format.
>
> **New Prompt (Revised Following Reviewer Suggestions):**
>
> In light of the reviewer’s concern and recent advances in prompting for multi-label classification (e.g., Choice Funnel Algorithm, Chen et al., 2025), we adopted a more structured, option-based prompt:
>
> > 任务：请分析以下语句，并判断其所反映的情绪。
> 语句：
>  {sentences}
> 可用情绪选项：
>  A. 喜悦
>  B. 悲伤
>  …
> 要求：请从“可用情绪选项”中选择所有语句中反映的情绪。
> 答案：
>
> English Translation:
> > Task: Please analyze the following sentence(s) and determine the emotions it expresses.
> Sentence(s):
> {sentences}
> Available emotion options:
> A. Joy
> B. Sadness
> …
> Instruction: Please select all emotions from the “Available emotion options” that are reflected in the sentence(s).
> Answer:
>
> This prompt restricts the output space to predefined options, reducing generation ambiguity and improving consistency across models.
>
> To further enhance reproducibility and reduce stochastic errors in multi-label prediction, we additionally adopted the Choice Funnel Algorithm as follows:
> 1. Add an extra option “None of the above”.
> 2. In each iteration, constrain the model to generate only the first token.
> 3. Select the option whose first token has the highest probability.
> 4. If the selected option is “None of the above” or its probability falls below a preset threshold, the process terminates.
> 5. Otherwise, save the selected emotion, remove it from the option list, and repeat.
> This procedure yields stable multi-label predictions without relying on free-form generation. Due to ongoing testing of the revised method, several models are still being tested, and while results were not fully finalized before the suggested rebuttal time, we will present the updated results as soon as they become available.
>
> This procedure yields stable multi-label predictions without relying on free-form generation. Below, we present results obtained using this more robust prompt design for multi-label classification:
> | Model Name            | Jaccard | Precision | Recall | Exact Match |
> |-----------------------|---------|-----------|--------|-------------|
> | Qwen2.5-72B-Instruct  | **0.454** | **0.413**   | **0.583** | **0.163**       |
> | Qwen2.5-7B-Instruct   | 0.344   | 0.368     | 0.410  | 0.149       |
> | Qwen3-8B              | 0.390   | 0.398     | 0.533  | 0.125       |
> | Qwen3-14B             | 0.396   | 0.374     | 0.507  | 0.123       |
> | Llama-3.3-70B-Instruct   | 0.397   | 0.385     | 0.547  | 0.108       |
> | gemma-3-27b-it        | 0.388   | 0.350     | 0.509  | 0.105       |
> | internlm3-8b-instruct    | 0.359   | 0.365     | 0.458  | 0.090       |
> | GLM-4-9B-Chat         | 0.262   | 0.366     | 0.369  | 0.051       |
>
> The new prompting method is more consistent than the previous one, and it shows that Qwen2.5-72B-Instruct consistently achieves the best performance across multiple complementary metrics. We will adopt this prompt and method in the revised PDF version.
>
> Reference:
> 1. Xu, W., Cui, S., Fang, X., Xue, C., Eckman, S., & Reddy, C. K. (2025). Sata-bench: Select all that apply benchmark for multiple choice questions. arXiv preprint arXiv:2506.00643.

---

### Official Review · Reviewer_ZFg7 · 2025-10-29

**Soundness:** 3
**Presentation:** 3
**Contribution:** 2
**Rating:** 6
**Confidence:** 3

**Summary:**

This paper introduced SinoMultiAffect (SMA) dataset which contains 4,500 Chinese sentences from social media, with 4,058 labeled for 35 fine-grained emotion categories. This dataset also includes functional magnetic resonance imaging (fMRI) recordings of the brain while human participants were reading the sampled sentences. This paper also introduced a VAD-guided human-LLM
alignment framework. Evaluations on this benchmark has been conducted across multiple models, such as Qwen3, Llama3.1 and Llama3.3 models.

**Strengths:**

- This paper introduced a novel dataset. It is a more comprehensive dataset compared to other Chinese emotion datasets
- This paper conducted detailed analyses across different dimensions of this dataset.

**Weaknesses:**

- Analyses over different models' performance on this dataset are limited (Table 3). It would be interested to see why some models are better than others for some metrics, but worse in other metrics.
- This paper only evaluates general instruct LLMs on this benchmark. However, emotion-related baselines also need to be evaluated on this constructed benchmark.

**Questions:**

- From Table 3, it is hard to tell which model achieves the best results among all models. What are the differences between these metrics?

---

> ### Author Response · Authors · 2025-11-22
> **General response**
>
> We sincerely thank Reviewer ZFg7 for the thoughtful and constructive feedback. Your comments greatly helped us clarify the analysis and strengthen the evaluation. In response to your suggestions, we have (i) re-organized and refined the descriptions of our evaluation metrics, highlighting their complementary roles in assessing different aspects of multi-label emotion prediction; (ii) updated our prompting method to ensure more consistent and reliable results across models; and (iii) conducted additional representation-level analyses to better characterize how different models’ internal emotion representations relate to their evaluation outcomes. We hope these revisions adequately address your concerns and contribute to a clearer and more rigorous presentation of our work.

---

> > ### Author Response · Authors · 2025-11-22
> > **Response to Weaknesses:**
> >
> > **Response to weakness 1 (about different model performance):**
> >
> > Thank you for this valuable suggestion. While it is challenging to determine the causal reasons behind why certain models outperform others on specific metrics given the time constraints of the rebuttal period, we conducted additional analyses to better characterize what kinds of models tend to perform better across different evaluation dimensions. To do this, we extracted the last-token embeddings of each sentence from all models, which provide a consistent representation of the model’s overall emotional interpretation. Using the human-annotated emotion with the highest intensity as a reference, we examined several representation-level properties of each model, including: (i) the distinctiveness of fine-grained emotion clusters, (ii) the level of structural alignment between model-derived and human emotion representations, and (iii) the predictive accuracy on dimensional ratings (valence, arousal, and dominance).
> >
> >
> > | Model Name              | Correlation with Human | Category Distinctiveness | Valence pred | Arousal pred | Dominance pred | VAD avg | ACC (Jaccard Similarity) |
> > |-------------------------|-------------------------|----------------------------|---------------|---------------|------------------|----------|----------------------------|
> > | Qwen2.5-72B-Instruct    | 0.572                   | **0.012**                  | **0.782**     | 0.447         | 0.204            | 0.478    | **0.454**                  |
> > | Qwen2.5-32B-Instruct    | 0.543                   | 0.010                      | 0.772         | **0.453**     | **0.220**        | **0.481**| 0.434                      |
> > | Qwen2.5-7B-Instruct     | 0.617                   | -0.025                     | 0.738         | 0.437         | 0.213            | 0.463    | 0.344                      |
> > | Qwen3-8B                | 0.569                   | -0.015                     | 0.730         | 0.447         | 0.205            | 0.461    | 0.390                      |
> > | Qwen3-14B               | 0.453                   | -0.026                     | 0.735         | 0.418         | 0.144            | 0.432    | 0.396                      |
> > | Llama-3.1-8B-Instruct   | 0.588                   | -0.026                     | 0.683         | 0.441         | 0.138            | 0.420    | 0.374                      |
> > | Llama-3.3-70B-Instruct  | 0.639                   | -0.003                     | 0.736         | 0.411         | 0.168            | 0.438    | 0.397                      |
> > | internlm3-8b-instruct   | **0.645**               | -0.006                     | 0.762         | 0.438         | 0.214            | 0.471    | 0.359                      |
> > | GLM-4-9B-Chat           | 0.619                   | -0.040                     | 0.730         | 0.436         | 0.186            | 0.451    | 0.262                      |
> >
> > Interestingly, we observe that models with higher category distinctiveness, that is, more separable fine-grained emotion clusters in their embedding space, tend to achieve higher Jaccard similarity on the multi-label prediction task. In contrast, greater similarity to human emotion-space structure (Correlation with Human) does not consistently correspond to better multi-label accuracy across models.
> >
> > While we cannot make strong causal claims, these patterns suggest that the ability to internally differentiate fine-grained emotional categories may be more directly beneficial for multi-label classification than global similarity to human-annotated structures. This may indicate that LLMs rely more on their own category-separation mechanisms than on human-like representational geometry when performing strict multi-label inference.
> >
> > **Response to weakness 2 (about evaluation of emotion-related baselines):**
> >
> > Thank you for this helpful suggestion. Following your comment, we searched for publicly available emotion-related baseline models that have been fine-tuned specifically for fine-grained emotion classification. We identified models such as SamLowe/roberta-base-go_emotions and arpanghoshal/EmoRoBERTa on HuggingFace; however, both are trained exclusively on English fine-grained emotion datasets and cannot be applied to our Chinese benchmark. Given this limitation, we focused our evaluation on open-source foundation LLMs in current study.

---

> > ### Author Response · Authors · 2025-11-22
> > **Response to Questions (about the differences among the metrics):**
> >
> > Thank you for raising this point. We agree that the previous presentation of various but overlapping metrics could be confusing. To address this, we have reorganized and clarified the evaluation metrics in the revised version. Specifically, we selected a set of complementary metrics that capture different aspects of multi-label prediction performance.
> >
> > - **Jaccard Index** evaluates the overall overlap between predicted and true labels.
> > - **Precision** measures how reliable the predictions are by quantifying how rarely the model produces incorrect emotions (low false positives).
> > - **Recall measures** how comprehensively the model captures underlying emotions in the text (low false negatives).
> > - **Exact Match** reflects strict correctness, requiring the predicted emotion set to match the ground truth exactly.
> >
> > These metrics thus capture different dimensions—strict accuracy, partial overlap, reliability, and coverage—and their combined use makes the relative strengths of different models clearer. More importantly, after we modified our prompting method, results became more consistant, which showed Qwen2.5-72B-Instruct consistently achieves the best performance in the most metrics.
> >
> > | Model Name              | Exact Match Accuracy | Jaccard Similarity | Precision | Recall |
> > |-------------------------|----------------------|----------------------|-----------|---------|
> > | Qwen2.5-72B-Instruct    | 0.163                | **0.454**            | **0.413** | **0.583** |
> > | Qwen2.5-32B-Instruct    | 0.147                | 0.434                | 0.382     | 0.574     |
> > | Qwen2.5-7B-Instruct     | 0.149                | 0.344                | 0.368     | 0.410     |
> > | Qwen3-8B                | 0.125                | 0.390                | 0.398     | 0.533     |
> > | Qwen3-14B               | 0.123                | 0.396                | 0.374     | 0.507     |
> > | Llama-3.1-8B-Instruct   | **0.219**            | 0.374                | 0.409     | 0.376     |
> > | Llama-3.3-70B-Instruct  | 0.108                | 0.397                | 0.385     | 0.547     |
> > | gemma-3-27b-it          | 0.105                | 0.388                | 0.350     | 0.509     |
> > | internlm3-8b-instruct   | 0.090                | 0.359                | 0.365     | 0.458     |
> > | GLM-4-9B-Chat           | 0.051                | 0.262                | 0.366     | 0.369     |

---

### Official Review · Reviewer_Zvhr · 2025-10-31

**Soundness:** 2
**Presentation:** 2
**Contribution:** 2
**Rating:** 4
**Confidence:** 4

**Summary:**

This paper introduces SinoMultiAffect (SMA), a new Chinese dataset that integrates fine-grained emotion annotations (35 categories + VAD + intensity) with neural data (fMRI) collected during reading tasks.
It contains 4,500 social-media sentences, of which 4,058 are annotated with multi-label emotions and dimensional ratings. A smaller subset (120 sentences) was presented to human participants under fMRI scanning to capture neural correlates of emotional language processing.

The authors benchmark 13 open-source LLMs on zero-shot Chinese emotion recognition, and propose a VAD-guided contrastive alignment model linking brain and text embeddings to study human–LLM alignment in affective representation.

**Strengths:**

Unique multimodal contribution. The first fine-grained Chinese emotional text dataset, accompanied by neuroimaging data, enables research that connects affective NLP, cultural linguistics, and cognitive neuroscience.

High-quality curation pipeline. Manual multi-label annotation, dimensional VAD scales, inter-rater analysis, and validation of emotion taxonomy demonstrate rigorous data design.

Cultural significance. Captures East-Asian emotion concepts (e.g., being moved, gratitude, healing), addressing the Western-centric bias of prior emotion corpora.

Solid baseline benchmarking. Evaluates 13 LLMs (8B–72B) under a consistent zero-shot protocol, providing a strong empirical reference for future studies.

**Weaknesses:**

1.  Extremely limited neural dataset
The fMRI component, though conceptually valuable, includes only three participants and 120 stimuli.
This scale is far below the threshold required to support statistically meaningful alignment claims, especially when it is the core contribution of the paper.
No within-subject or cross-subject validation, no voxel-wise encoding/decoding models, and no statistical tests (e.g., permutation or bootstrap confidence intervals) are reported.
Given the known noise level of fMRI (hemodynamic delay, low SNR, inter-subject variability), the presented alignment metrics (Hit@1≈0.17, Hit@3≈0.37) are effectively at or slightly above random, and cannot substantiate robust “brain–LLM alignment.”
Consequently, the neural findings should be treated as proof-of-concept, not evidence of genuine representational correspondence.

2. Experimental setting concerns.
The paper conflates two distinct settings, but the authors didn't explicitly mention this. The text-only emotion classification is genuinely zero-shot (LLM inference without training), but the fMRI–LLM alignment still trains two projection MLPs on paired data with a supervised contrastive objective. Therefore, this mapping does not constitute zero-shot recognition of neural signals.
Readers may incorrectly infer that LLMs intrinsically interpret fMRI patterns, when in fact a learned mapping is mediating the correspondence. This conceptual overreach undermines the scientific precision of the claims and should be explicitly corrected. In this regard, if the zero-shot setting of the LLM is conducted on the text-only emotion classification, the technical contribution will be extremely limited.

3. Limited methodological novelty
The VAD-guided alignment extends prior contrastive multimodal frameworks (e.g., CLIP-style InfoNCE) by adding auxiliary regression losses for emotion dimensions. While this is reasonable, it is incremental rather than groundbreaking. There is no new theoretical insight into multimodal representation learning, nor an analysis of how VAD supervision alters embedding geometry beyond sensitivity/entropy plots. The work contributes more as a dataset paper than as a methodological advance for ICLR’s main track.

4. Over-interpretation of weak correlations
The authors report modest improvements when injecting emotion features (e.g., Brain-Only variant Hit@1=0.169 vs. 0.082 baseline), but the absolute performance remains low and unvalidated.
Without randomization tests or comparisons to trivial baselines (e.g., linear regression decoding, canonical correlation analysis, RSA), it is impossible to determine whether these results reflect genuine affective structure or dataset noise.
The qualitative statements about “strong utility of affective cues” and “alignment within a shared emotional space” are not quantitatively supported.

**Questions:**

The authors can address the concerns from the following points:
Clarify and separate zero-shot vs. trained alignment claims.
Increase the fMRI participant/sample size or clearly present it as exploratory.
Add statistical tests.
Strengthen discussion and contribution of how this dataset can drive new ML directions (e.g., emotion-grounded representation learning).

---

> ### Author Response · Authors · 2025-11-22
> **General response**
>
> We sincerely thank reviewer Zvhr for the constructive comments and suggestions. We understand the reviewer’s concerns regarding the methodological novelty, and in our response we have placed clearer emphasis on the value and contribution of the dataset itself, which is the primary focus of this work. We have also clarified the distinction between the two experimental settings as recommended, to avoid potential misinterpretation of the alignment component. We hope that these revisions and explanations can address the reviewer’s concerns about the contribution of the study.

---

> > ### Author Response · Authors · 2025-11-22
> > **Response to Weaknesses:**
> >
> > About methodological novelty:
> >
> > Thank you for the reviewer’s assessment. We agree that our work does not introduce a new theoretical multimodal learning method. The VAD-guided alignment is intended as a proof-of-concept demonstration of the usefulness of our dataset rather than a methodological contribution in itself.
> >
> > Our primary goal is to provide a carefully curated, culturally grounded Chinese dataset that integrates (i) fine-grained multi-label emotional categories, (ii) continuous VAD dimensions, and (iii) fMRI recordings collected during naturalistic emotional language reading . Existing Chinese resources either contain only coarse categories, provide single-label annotations, or lack neural data. There is currently no fine-grained Chinese emotion dataset with both multi-label structure and accompanying fMRI recordings, making SinoMultiAffect a resource that fills a clear gap in the field.
> >
> > The alignment model is therefore used only to illustrate one possible way the dataset enables multimodal research, not to claim methodological novelty. We will clarify this motivation and adjust the framing so that the dataset contribution is primary and the alignment serves as an example application.

---

> > ### Author Response · Authors · 2025-11-22
> > **Response to Questions:**
> >
> > **About clarifying zero-shot vs. trained alignment claims:**
> >
> > To address this concern, we will explicitly clarify in the revised manuscript that our work contains two distinct experimental settings in Section 5:
> > - a text-only multi-label emotion classification task, where LLMs are evaluated in a pure zero-shot setting without any gradient updates; and
> > - a fMRI-LLM alignment task, where we trained two projection MLPs on paired brain-text data with a supervised contrastive objective (Eq. 1-3, Fig. 5).
> >
> > We acknowledge that the current draft does not state this distinction explicitly enough and might give the impression that LLMs "intrinsically" decode fMRI signals. In the revision, we will clearly label the second part as a supervised brain–text alignment analysis rather than a "zero-shot" setting, and describe it as an exploratory investigation aimed at probing whether emotional information can facilitate the mapping between fMRI responses and text-derived representations.
> >
> > **About the sample size of fmri:**
> >
> > We understand the reviewer’s concern regarding the small sample size of the fMRI dataset. We fully agree that with only three participants and 120 stimuli, the current scale is insufficient to support strong statistical claims. In response, we have revised the relevant descriptions throughout the manuscript to present the fMRI-based alignment analysis as an exploratory, proof-of-concept study rather than conclusive evidence of brain–LLM representational correspondence.
> >
> > **About adding statistical tests:**
> >
> > For the brain-LLM alignment part, we did multiple seeds (seed = 1,3,5,742,100) for a more robust result. Following your suggestion, we have added a statistical test for interpretation of results.  First, we ran the alignment experiments across multiple random seeds (1, 3, 5, 7, 42, 100) to reduce variance and obtain more stable estimates. Second, following your suggestion, we added formal statistical tests on the retrieval accuracies (Hit@1 and Hit@3). Specifically, we conducted an ANOVA across all evaluated models (the Basic model and variants incorporating emotional information). The results revealed a significant overall model effect (F(4,25) = 6.622, p < 0.001). Post-hoc comparisons further showed that the BRAIN_ONLY model not only achieved the highest retrieval performance but also significantly outperformed the Basic model in both Hit@1 (p_adj = 0.017) and Hit@3 (p_adj = 0.001). We will report these statistics in the updated PDF version.
> >
> > Furthermore, to ensure that the improvements are not only numerical but also reflected in representational geometry, we added an interpretability analysis examining how well brain and text embeddings organize along affective dimensions. Specifically, we computed Spatial-VAD correlation, defined as the Spearman correlation between pairwise Euclidean distances in the embedding space and pairwise valence, arousal, and dominance differences. As shown in the following table, the BRAIN_ONLY model shows substantial increases across all three affective dimensions, with especially large gains in Arousal (Δ = +0.3412 for brain embeddings and Δ = +0.2434 for text embeddings) (see Figure S7 in appendix E.2).
> >
> > Brain Embedding Performance (Spatial–VAD Correlation)
> > | Dimension | Basic  | Pred-only-drop | Delta   |
> > | --------- | ------ | -------------- | ------- |
> > | Valence   | 0.0228 | 0.2112         | +0.1884 |
> > | Arousal   | 0.0076 | 0.3489         | **+0.3412** |
> > | Dominance | 0.0322 | 0.3183         | +0.2861 |
> >
> > Text Embedding Performance (Spatial–VAD Correlation)
> > | Dimension | Basic  | Pred-only-drop | Delta   |
> > | --------- | ------ | -------------- | ------- |
> > | Valence   | 0.0279 | 0.1440         | +0.1160 |
> > | Arousal   | 0.0127 | 0.2562         | **+0.2434** |
> > | Dominance | 0.0602 | 0.2954         | +0.2352 |

---

> > ### Author Response · Authors · 2025-11-22
> > **Response to the contribution of the dataset:**
> >
> > Thank you for the suggestion. We address the potential contributions as follows:
> >
> > Our dataset provides both academic and practical value. It enables future research on emotion representation in language and the brain, and can be directly applied to the development of large language models and embodied agents with emotion-related capabilities, particularly for recognizing and understanding nuanced emotional expressions in Chinese. Importantly, our goal is to isolate emotion signals expressed purely through text, as this is the modality through which large language models are most commonly accessed and used today.
> >
> > Beyond categorical annotations, SinoMultiAffect offers emotion intensity scores together with continuous ratings of valence, arousal, and dominance, supporting both discrete and dimensional perspectives of affect. This dual-level design enables analyses that go beyond coarse categorical distinctions to capture the subtleties of emotional expression in language. In addition, by providing multiple human labels that preserve both consensus and the natural variability of fine-grained emotional judgments, our dataset reflects how humans genuinely interpret nuanced emotions rather than collapsing them into overly simplified categories.
> >
> > In summary, our goal is to provide a human-labeled resource that retains the full complexity of emotional meaning. We believe that this fine-grained structure supplies richer supervision signals for machine learning models, enabling future work on emotion-grounded representation learning, more culturally aligned affective modeling, and LLMs that can better understand, differentiate, and support human emotions in real-world interactions. We anticipate that SinoMultiAffect will foster both theoretical advances and practical applications in emotion-centered artificial intelligence.
> >
> > We will strengthen the Discussion part in the updated PDF version.

---

### Official Review · Reviewer_KKFD · 2025-11-04

**Soundness:** 3
**Presentation:** 3
**Contribution:** 2
**Rating:** 4
**Confidence:** 3

**Summary:**

The paper introduces SinoMultiAffect (SMA), a novel Chinese emotion dataset. Its primary contribution is the dataset itself: 4,058 Chinese social media texts annotated with a four-part structure: (1) a fine-grained, multi-label taxonomy of 35 emotion categories (2) continuous dimensional ratings for Valence, Arousal, and Dominance (VAD) (3) emotion intensity scores and (4) corresponding fMRI data from human participants reading the texts. The paper validates this resourceby providing a comprehensive zero-shot benchmark of 13 large language models (LLMs) on the emotion recognition task and by proposing a VAD-guided framework to align LLM embeddings with the fMRI data.

**Strengths:**

Reason to Accept

- Novel Multi-Modal Dataset: This is the first Chinese dataset to integrate fine-grained (35 labels), multi-label text with both dimensional (VAD) ratings and neural (fMRI) data, which is a unique resource for interdisciplinary research.
- Addresses Critical Gaps in the Field: The dataset directly addresses the need for culturally specific, non-English emotion resources. It provides far greater label granularity (35 labels) than the largest existing Chinese dataset, CMACD (6 labels) , and adds neural/VAD data, which are missing from most text-based resources.
- Strong LLM Benchmark: The paper provides a solid and immediately useful zero-shot benchmark of 13 modern LLMs. This validates the dataset's utility for fine-grained NLP tasks and provides a strong baseline for future models

**Weaknesses:**

Reasons to Reject


- Low Interrater Agreement (IAA): The paper reports an average Jaccard index of 0.346 for label agreement, which is state as moderate. This score is low, indicating that the 35-label taxonomy is likely too ambiguous or subjective. This low reliability calls the validity/quality of the dataset's ground truth labels.
- LLM Bias: The paper claims LLMs show a systematic overprediction bias. This may be a misinterpretation. Given the low human IAA (0.346) , it is more likely the LLMs are correctly identifying the same label ambiguity that individual human raters saw, and the ground truth is an artificial product of the 2/3 consensus mechanism filtering this ambiguity out.
- N=3 for fMRI data is too low to obtain statistically meaningful results.

**Questions:**

Questions

See above in Reasons to Reject. Additionally

- Your correlation analysis (Figure 3) shows high similarity between certain emotion pairs (e.g., 'remorse'/'guilt'). Combined with the low IAA , do you believe all 35 categories are truly distinct and reliably identifiable from text?
- The limitations section notes the dataset's scale. Are there concrete plans to collect fMRI data from a statistically significant sample (e.g., N > 30) to properly validate the VAD-guided alignment framework?
- Given the low IAA , do you plan to conduct another annotation round? Will you consider collapsing similar categories into a smaller, more robust label set?
- Instead of a 2/3 consensus , have you considered releasing the raw rater counts (e.g., 1/3, 2/3, 3/3) as soft labels? This would allow models to learn from the human-level ambiguity you identified, rather than training on a filtered and potentially biased ground truth.

---

> ### Author Response · Authors · 2025-11-21
> **General response**
>
> We thank Reviewer KKFD for the careful and thoughtful review, as well as insightful comments. We fully understand the reviewer’s concern regarding the credibility of overly fine-grained emotion categories. In response, we provided additional analyses and compared our results with findings from a highly cited prior fine-grained dataset study to demonstrate consistency (Resonse to Weakness 1, Response to Question 1).  We also explained the rationale behind maintaining granularity from both psychological and modeling perspectives (Response to Question 3). In addition, for Weakness 2, we refined our analysis by comparing LLM outputs with all human-provided labels and updated our wording accordingly to more accurately characterize the phenomenon (Response to Weakness 2). We hope that these clarifications are able to address the reviewer’s concerns.

---

> > ### Author Response · Authors · 2025-11-21
> > **Response to Weakness 1 (about low IAA):**
> >
> > Thank you for highlighting this issue. The reported average Jaccard index of 0.346 was across all 4,500 annotated texts, including those items for which annotators did not reach consensus. When focusing only on the 4,058 texts with consensus labels, the average Jaccard index increases to 0.38. The reason for this relatively low IAA may be due to the fact that our annotation protocol explicitly allowed and even encouraged annotators to select two emotion labels for each sentence when appropriate. Under a 35-category fine-grained taxonomy, such a multi-label setup naturally leads to relatively lower Jaccard scores.
> >
> > More importantly, Jaccard index alone may underestimate agreement for multi-label, fine-grained emotions. When examining interrater correlations across the 35 emotion dimensions, the agreement level is comparable to that reported in GoEmotions (28 catergories, see following Table 1 and 2), a well-established English fine-grained emotion dataset (Demszky, 2020). This suggests that although fine-grained emotion labeling is inherently subjective, the reliability of our annotations aligns with the accepted standard in the field.
> > We have provided the sentence-level Jaccard index for every item in the new supplementary materials (see File: dataset_SinoMultiAffect/samples_consensus_4058.csv). This allows users of the dataset to identify which texts exhibit higher annotator consensus (i.e., more certain emotional signals) and which ones show greater variability in emotional interpretation. Such information can help researchers tailor their modeling choices and better understand the ambiguity structure within the dataset. To transparently present this variability, we provide the full distribution of sentence-level Jaccard scores in Figure S1 (Appendix B).
> >
> > Table 1: Interrater agreement measured by correlation in Goemotions (Demszky, 2020).
> > | Emotion        | Interrater Correlation |
> > |----------------|------------------------|
> > | admiration     | 0.535                  |
> > | amusement      | 0.482                  |
> > | anger          | 0.207                  |
> > | annoyance      | 0.193                  |
> > | approval       | 0.385                  |
> > | caring         | 0.237                  |
> > | confusion      | 0.217                  |
> > | curiosity      | 0.418                  |
> > | desire         | 0.177                  |
> > | disappointment | 0.186                  |
> > | disapproval    | 0.274                  |
> > | disgust        | 0.192                  |
> > | embarrassment  | 0.177                  |
> > | excitement     | 0.193                  |
> > | fear           | 0.266                  |
> > | gratitude      | 0.645                  |
> > | grief          | 0.162                  |
> > | joy            | 0.296                  |
> > | love           | 0.446                  |
> > | nervousness    | 0.164                  |
> > | optimism       | 0.322                  |
> > | pride          | 0.163                  |
> > | realization    | 0.194                  |
> > | relief         | 0.172                  |
> > | remorse        | 0.178                  |
> > | sadness        | 0.346                  |
> > | surprise       | 0.275                  |
> >
> > Table 2: Interrater agreement measured by correlation in Ours
> > | Emotion           | Interrater Correlation |
> > |------------------|------------------------|
> > | disappointment     | 0.6824 |
> > | appreciation       | 0.6480 |
> > | confusion           | 0.5669 |
> > | gratitude          | 0.5652 |
> > | guilt               | 0.5617 |
> > | looking forward     | 0.5457 |
> > | nostalgia           | 0.5152 |
> > | optimism            | 0.5019 |
> > | disgust                 | 0.4981 |
> > | admiration              | 0.4867 |
> > | sadness                 | 0.4845 |
> > | surprise                | 0.4803 |
> > | joy                     | 0.4610 |
> > | fear                    | 0.4559 |
> > | anger                   | 0.4550 |
> > | anxiety                 | 0.4236 |
> > | sympathy                | 0.3971 |
> > | neutral                 | 0.3859 |
> > | remorse                 | 0.3847 |
> > | curiosity               | 0.3742 |
> > | being moved             | 0.3703 |
> > | pity                    | 0.3580 |
> > | love                    | 0.3337 |
> > | hesitant                | 0.3289 |
> > | healing                 | 0.3032 |
> > | embarrassment           | 0.2941 |
> > | satisfaction            | 0.2925 |
> > | being at ease           | 0.2877 |
> > | pride in oneself        | 0.2863 |
> > | pessimism               | 0.2754 |
> > | cynicism                | 0.1993 |
> > | alarm                   | 0.1907 |
> > | trust                   | 0.1641 |
> >
> > Reference:
> > 1. Demszky, D., Movshovitz-Attias, D., Ko, J., Cowen, A., Nemade, G., & Ravi, S. (2020). GoEmotions: A dataset of fine-grained emotions. arXiv preprint arXiv:2005.00547. (citation count: 1194)

---

> > ### Author Response · Authors · 2025-11-21
> > **Response to Weakness 3 and Question 2 (about insufficient fmri samples):**
> >
> > We appreciate the reviewer’s concern regarding the small sample size of the fMRI data. We agree that N = 3 is insufficient for drawing population-level statistical conclusions, and we acknowledge this as a limitation. Our primary goal in this work, however, is to introduce a nuanced, fine-grained emotional text dataset, and the accompanying fMRI recordings are intended as an exploratory, proof-of-concept component rather than a large-scale neuroimaging study. They serve as an initial step toward leveraging neural signals as a biologically grounded supervision source, which may help future models better align with human emotional representations. We will further expand the fMRI sample size using our established pipeline to support more robust and population-level analyses in future work.

---

> > ### Author Response · Authors · 2025-11-21
> > **Response to Questions 1 (about high similarity in emotion categories):**
> >
> > Thank you for raising this important point. In addition to what we mentioned in our response to weakness 1, we also conducted a Pearson correlation analysis between all emotion pairs. The resulting correlation range (-0.31 to 0.36) is highly comparable to that reported in prior fine-grained English datasets such as GoEmotions, which contains 28 categories and exhibits a similar correlation range of approximately -0.3 to 0.3. Given that our taxonomy includes more categories (35 vs. 28), a slightly wider correlation spread is expected and remains within a reasonable range of fine-grained affective concepts.
> >
> > Furthermore, we would like to highlight that the presence of semantically related but distinct emotion categories is typical in fine-grained emotion taxonomies and can enrich the understanding of subtle affective differences. Psychological studies consistently show that the ability to use and discriminate a rich set of emotion words, referred to as emotional granularity, is associated with better emotion regulation and mental well-being (Kashdan et al., 2015). From this perspective, maintaining fine-grained distinctions is not only theoretically meaningful but also practically valuable. For example, in Chinese, remorse and guilt both involve negative self-evaluative emotions, but they differ in their core psychological mechanisms. Remorse arises mainly from regretting a poor decision or a missed opportunity; it is outcome-focused and reflects a counterfactual comparison like “things would have been better if I had chosen differently.” In contrast, guilt carries a stronger moral component and emerges when one recognizes having harmed someone else or violated personal moral standards.
> >
> > In summary, our goal is to provide a human-labeled dataset that preserves both the consensus and the natural variability of nuanced emotional expressions. We believe that retaining this fine-grained structure is essential for developing LLMs that can better understand, differentiate, and support human emotions in real-world interactions. Such detailed annotations offer richer supervision signals, enabling future models to capture subtle emotional distinctions that are crucial for effective human–AI emotional communication.

---

> > ### Author Response · Authors · 2025-11-21
> > **Response to Questions 3 and 4:**
> >
> > **Question 3 (about conducting another annotation round):**
> >
> > Thank you for your suggestions. We do plan to further expand the size of our dataset. For sentences with particularly low Jaccard index scores (e.g., < 0.2; 1,043 out of 4,058), we will include additional annotators in the next round of data collection to better assess category consensus and reduce uncertainty where possible.
> >
> > Regarding the label taxonomy, we believe that preserving the full set of categories is important for supporting research on fine-grained emotional understanding. Although merging categories might improve agreement scores, it would also reduce the semantic resolution necessary for modeling and examining subtle emotional distinctions. Moreover, ambiguity in emotional interpretation is not merely noise; it is an inherent property of real-world emotional expression. Some texts genuinely allow for multiple plausible emotional readings, and capturing this variability, is important for both psychological validity and for developing models capable of handling fine-grained emotion reasoning.
> >
> > Although we aim to preserve the original fine-grained structure, we appreciate the reviewer’s insightful suggestion and have accordingly added a complementary, relatively coarse-grained categorization. Specifically, we grouped the 35 fine-grained emotions into 17 broader categories based on an established prior emotion ontology (Xu et al., 2008, see Table 3 in the following). This provides a higher-level organizational structure that users may flexibly adopt for different research needs, while still maintaining the fine-grained distinctions that motivated our dataset design. Importantly, this hierarchical labeling scheme enhances usability without compromising granularity. Under the new coarse taxonomy, the Jaccard index increases to 0.486 across all 4,058 samples.
> >
> > Table 3: Hierarchical emotion taxonomy mapping 35 fine-grained categories to 17 coarse-grained classes.
> > | **English Superclass** | **English Subclass**            |
> > |------------------------|---------------------------------|
> > | Joy                    | Joy                             |
> > | Joy                    | Self-satisfaction               |
> > | Joy                    | Optimism                        |
> > | Joy                    | Satisfaction                    |
> > | Joy                    | being at ease      |
> > | Joy                    | Healing             |
> > | Affection              | Love                            |
> > | Affection              | Appreciation                    |
> > | Affection              | Being Moved                     |
> > | Affection              | Admiration                      |
> > | Affection              | Sympathy                        |
> > | Praise                 | Gratitude                       |
> > | Trust                  | Trust                           |
> > | Wishing                | Looking Forward                 |
> > | Anger                  | Anger                           |
> > | Anger                  | Cynicism                        |
> > | Sadness                | Sadness                         |
> > | Sadness                | Pessimism                       |
> > | Disappointment         | Disappointment                  |
> > | Disappointment         | Pity                            |
> > | Guilt                  | Remorse                         |
> > | Guilt                  | Guilt                           |
> > | Longing                | Nostalgia                       |
> > | Fear                   | Fear                            |
> > | Fear                   | Alertness                       |
> > | Shyness                | Shyness                         |
> > | Annoyance              | Anxiety                         |
> > | Annoyance              | Embarrassment                   |
> > | Annoyance              | Confusion                       |
> > | Disgust                | Disgust                         |
> > | Criticism              | Hesitant                        |
> > | Envy                   | Envy                            |
> > | Surprise               | Surprise                        |
> > | Surprise               | Curiosity                       |
> >
> > Reference:
> > 1. Xu, L., Lin, H., Pan, Y., et al. (2008). Construction of an emotional lexicon ontology. Journal of the China Society for Scientific and Technical Information, 27(2), 180–185.
> >
> >
> > **Question 4 (about providing soft labels):**
> >
> > We appreciate this insightful suggestion. Indeed, releasing the raw rater distributions can provide valuable information about human-level ambiguity and allow models to learn from soft labels rather than relying solely on a filtered consensus. In the updated supplementary materials, we included the full rater count for each sample (labels of 1/3, 2/3, and 3/3 agreement, see Supplementary File: dataset_SinoMultiAffect/samples_consensus_4058.csv). This will enable researchers to incorporate rater uncertainty directly into model training or evaluation.

---

> > ### Author Response · Authors · 2025-11-22
> > **Response to Weakness 2 (revision to the "LLM Bias" phrasing):**
> >
> > Thank you for this insightful comment. We agree that our earlier phrasing, “LLMs show a systematic overprediction bias”, was not a precise way to describe the phenomenon. We appreciate the reviewer’s perspective that some of these cases may reflect genuine ambiguity in emotional interpretation rather than a model bias.
> >
> > To more accurately characterize this phenomenon, we have added an additional analysis comparing LLM predictions with all human-provided labels, rather than relying only on consensus labels (see the following table). In line with this clarification, we have removed the earlier “overprediction” phrasing from the manuscript to avoid potential misinterpretation. The revised analysis allows us to capture the full distribution of human emotional interpretations and to examine how closely LLM outputs align with the range of emotions identified by annotators in ambiguous cases. We hope that this expanded perspective provides new insights into how LLMs navigate and represent the variability inherent in fine-grained emotion recognition. We will add this result in the appendix part in the revised PDF version.
> >
> > | Model Name            | Jaccard_all | Jaccard_consensus | Precision_all | Precision_consensus | Recall_all | Recall_consensus | ExactMatch_all | ExactMatch_consensus |
> > |-----------------------|-------------|--------------------|----------------|----------------------|-------------|--------------------|-----------------|------------------------|
> > | **Qwen2.5-72B-Instruct** | **0.409**    | **0.454**          | **0.640**       | **0.413**             | **0.329**    | **0.583**          | **0.066**        | **0.163**               |
> > | Qwen2.5-7B-Instruct   | 0.279       | 0.344              | 0.552          | 0.368                | 0.246       | 0.410              | 0.024          | 0.149                  |
> > | Qwen3-8B              | 0.351       | 0.390              | 0.607          | 0.398                | 0.306       | 0.533              | 0.045          | 0.125                  |
> > | Qwen3-14B             | 0.352       | 0.396              | 0.578          | 0.374                | 0.269       | 0.507              | 0.044          | 0.123                  |
> > | Llama-3.3-70B-Instruct| 0.367       | 0.397              | 0.590          | 0.385                | 0.323       | 0.547              | 0.048          | 0.108                  |
> > | gemma-3-27b-it        | 0.388       | 0.388              | 0.558          | 0.350                | 0.325       | 0.509              | 0.054          | 0.105                  |
> > | internlm3-8b-instruct | 0.333       | 0.359              | 0.541          | 0.365                | 0.283       | 0.458              | 0.038          | 0.090                  |
> > | GLM-4-9B-Chat         | 0.282       | 0.262              | 0.516          | 0.366                | 0.250       | 0.369              | 0.028          | 0.051                  |

---

### Author Response · Authors · 2025-11-29
**Summary of responses**

We sincerely thank Area Chairs, Program Chairs and all reviewers for their time, careful evaluation, and constructive feedback on our work. In this study, we introduced **SinoMultiAffect**, a Chinese multimodal dataset that integrates fine-grained emotion annotations (35 nuanced categories) with fMRI recordings, enabling a proof-of-concept exploration of how language-based emotion representations in large language models align with neural responses when emotion information is incorporated as a supervised training signal.

We have responsed to reviewers' comments through the following points:
- We argued about the necessity of keeping these nuanced categories based on psychological evidence. Although some concerns were raised about potential semantic overlap among certain emotions, we believe that retaining this fine-grained structure is essential for developing LLMs that can better understand, differentiate, and support human emotions in real-world interactions.
    - Response to reviewer KKFD: Response to Questions 3 and 4]
- We referenced prior fine-grained emotion datasets in English and showed that their inter-rater agreement indices are comparable to ours, thereby addressing Reviewer KKFD’s concern regarding the seemingly low IAA.
    - Response to reviewer KKFD: Response to Weakness 1 (about low IAA)
    - Response to Questions 1 (about high similarity in emotion categories)]
- We addressed the concern of reviewer Zvhr about methodological novelty by clarifying that the central contribution of this work lies in the value of the dataset itself.
    - Response to reviewer Zvhr: Response to Weaknesses
    - Response to reviewer Zvhr: Response to the contribution of the dataset
- In the evaluation part, we first applied a more robust prompt and algorithm for multi-label tasks, and then reorganised the metrics of evaluating the fine-grained emotion recognition ability of LLM, making them more clear to understand their performance from different perspectives. Revision can be found in the 5.1.1 and 5.1.2 parts of the revised PDF version.
    - Response to reviewer 92LJ: Response to Weakness 5 and Question 1 (about explicit prompt information)
    - Response to reviewer ZFg7: Response to Questions (about the differences among the metrics)]
- For the fMRI modality, we clarified that it serves as a proof-of-concept given the challenges of large-scale neural data collection. We also argued for its value in enabling future research aimed at improving LLMs through neural supervision.
    - Response to reviewer KKFD: Response to Weakness 3 and Question 2 (about insufficient fmri samples)
    - Response to reviewer Zvhr: Response to Questions (About the sample size of fmri)

Our dataset provides both academic and practical value. It enables future research on emotion representation in language and the brain, and can be directly applied to the training of large language models and embodied agents with emotion-related capabilities, particularly for recognizing and understanding nuanced emotions in Chinese.

We have revised our paper according to the reviewers’ suggestions, and we would also like to thank the Area Chair once again for time and careful assessment of this work.

---

### Meta-Review · Area_Chair_ezYx · 2025-12-05

**Summary:**

The paper proposes SinoMultiAffect, a Chinese multi-model (text+fMRI) dataset for fine-grained emotional analysis, where the main task is to predict multi-label emotions. The authors also showed an analysis on VAD-guided human--LLM alignment.

**Reviewer Concerns:**

Reviewers mainly challenged the quality of the dataset, including the size and inter-annotator correlation.

Reviewers also raised concerns about the significance of the dataset and its analysis. I concur with this concern, as it's not clear to me how practical is the task: obtaining fMRI data is highly unlikely for daily use. The authors may claim that the dataset is mainly for scientific exploration (not for application), but then I would like to see the actual interesting findings coming out of the dataset (not leaving it to other people). Currently, the VAD-guided human--LLM alignment is rudimentary and insignificant, as the authors admit that it's "proof-of-concept".

**Reviewer Scores:**

Many (although not all) reviews are highly AI-flavored. They have template-based structures, with many long and confusing expressions. Such reviews and scores are not helpful.

Note: The submitted manuscript is also AI-flavored. I acknowledge that the authors may have come with all substantial ideas by themselves, the manuscript has many long and sophisticated sentences/expressions, which makes it slow to precisely understand the meaning.

The significance of the work cannot be addressed by author response or revisions.

---

### Decision · Program_Chairs · 2026-01-26

Reject